# Differential Privacy Guarantees of Markov Chain Monte Carlo Algorithms

**Andrea Bertazzi** [1]  **Tim Johnston** [2]  **Gareth O. Roberts** [3]  **Alain Durmus** [1]

## Abstract

This paper aims to provide differential privacy (DP) guarantees for Markov chain Monte Carlo (MCMC) algorithms. In a first part, we establish DP guarantees on samples output by MCMC algorithms as well as Monte Carlo estimators associated with these methods under assumptions on the convergence properties of the underlying Markov chain. In particular, our results highlight the critical condition of ensuring the target distribution is differentially private itself. In a second part, we specialise our analysis to the unadjusted Langevin algorithm and stochastic gradient Langevin dynamics and establish guarantees on their (Rényi) DP. To this end, we develop a novel methodology based on Girsanov's theorem combined with a perturbation trick to obtain bounds for an unbounded domain and in a non-convex setting. We establish: (i) uniform in $n$ privacy guarantees when the state of the chain after $n$ iterations is released, (ii) bounds on the privacy of the entire chain trajectory. These findings provide concrete guidelines for privacy-preserving MCMC.

## 1. Introduction

The framework of differential privacy (DP) (Dwork, 2006; Dwork et al., 2006) has become the standard approach for designing statistical and machine learning algorithms with quantitative guarantees on the information that their output reveals about the data. In particular, the DP guarantees of Bayesian statistical methods have already been investigated in various works, typically focusing on the DP analysis of samples drawn from the posterior distribution associated with the data and problem at hand (Wang et al., 2015; Dimitrakakis et al., 2017; Geumlek et al., 2017; Hu et al., 2025). In practice, Markov chain Monte Carlo (MCMC) algorithms

are required to produce approximate samples from the posterior, and for this reason their DP guarantees have also been the object of extensive research (Heikkilä et al., 2019; Yıldırım & Ermics, 2019; Li et al., 2019; Chourasia et al., 2021; Altschuler & Talwar, 2022; Zhang & Zhang, 2023).

Several works obtain DP of the one-step transitions of the MCMC algorithm by injecting extra noise in various ways, e.g. in the acceptance-rejection step of Metropolis-Hastings-type algorithms (Heikkilä et al., 2019; Yıldırım & Ermics, 2019), or leveraging the additional randomness already introduced by subsampling strategies (Wang et al., 2015; Bierkens & Duncan, 2022), optionally together with clipping of the gradient that drives the moves of the chain (Song et al., 2013; Abadi et al., 2016). Many iterations of the MCMC chain are required to approach the posterior distribution, and the typical strategy to obtain guarantees on the DP of the algorithm after some number $n$ of iterations is to use composition bounds (Abadi et al., 2016; Wang et al., 2019; Bassily et al., 2014; Ganesh et al., 2023). However, the privacy implied using composition bounds for Markov chains naturally decays with the number of steps (Kairouz et al., 2015) and as a consequence the injected noise must scale with the number of iterations in order to obtain uniform privacy bounds for the final draw. Many other works using different techniques to prove uniform-in-time privacy bounds for the final draw from a Markov chain have bad dependence on the radius of the state space or the stepsize, see (Altschuler & Talwar, 2022; Asoodeh & Diaz, 2023). These limitations were overcome in Chourasia et al. (2021), in which uniform-in-time bounds on the (Rényi) DP of gradient descent-type algorithms was obtained, however under a strong convexity assumption.

**Contributions of this work**    In this paper we study the differential privacy of MCMC algorithms. The paper is divided in two parts.

In the first part (Section 3) we consider general MCMC algorithms and establish clear connections between the DP guarantees of the posterior distribution and of a corresponding MCMC algorithm. Assuming DP of the posterior, Propositions 3.2 and 3.10 show $(\varepsilon, \delta)$-DP for the MCMC algorithm when we either release the $n$-th state of the chain or a (noisy) Monte Carlo estimator. These result show that, under suitable assumptions, the MCMC algorithm inherits the "good"

[1]École polytechnique, Institut Polytechnique de Paris, France [2]Université Paris Dauphine - PSL, France [3]University of Warwick, UK. Correspondence to: Andrea Bertazzi <andrea.bertazzi@polytechnique.edu>.

*Proceedings of the 42^{nd} International Conference on Machine Learning*, Vancouver, Canada. PMLR 267, 2025. Copyright 2025 by the author(s).

privacy properties of the posterior. Analogously, Propositions 3.3 and 3.4 illustrate how "bad" privacy properties of the posterior affect the privacy of an MCMC algorithm. In particular, Proposition 3.3 shows how an MCMC algorithm violates DP at a certain level $(\varepsilon, \delta)$ whenever it is sufficiently close to the posterior, assuming the posterior is not $(\varepsilon, \delta')$-DP for some $\delta' > \delta$. On the other hand, Proposition 3.4 shows that if the posterior has weaker DP guarantees than the MCMC algorithm, then the law of the MCMC chain after $n$ iterations can be far from the posterior in total variation distance. These results emphasise the importance of starting with a differentially private posterior, rather than only focusing on the one-step DP of the MCMC algorithm as often suggested in the literature. This viewpoint guarantees that the output of the MCMC algorithm can then be both private and close to the posterior.

In Section 4 we prove privacy bounds for a class of Markov chains in a non-convex setting, using a novel approach to establish uniform-in-time $(\varepsilon, \delta)$-DP and Rényi-DP guarantees of the $n$th state of the chain, as well as bounds for the trajectory up to the $n$th state. We show in Section 4.1 that it is possible to prove privacy guarantees via high probability bounds on the Radon-Nikodym derivative of different probability measures on the underlying probability space. In Section 4.2 we illustrate how this strategy can be applied to the case of MCMC algorithms based on diffusions, where we rely on Girsanov's theorem to express the Radon-Nikodym derivative. Using a careful perturbation technique, we are able to apply our approach to the case where only the final state after $n$ iterations is released, and not the entire path. In this case, we obtain uniform-in-time guarantees both for DP and Rényi-DP. Our strategy of proof also allows to obtain bounds for the entire trajectory of a variety of stochastic algorithms, improving on $(\epsilon, \delta)$-DP composition bounds and matching Rényi composition bounds. In particular, we show that the entire trajectory up to the $n$-th iteration is $(\varepsilon, \delta)$-DP, where $\varepsilon$ is $O(n + \sqrt{n \log(1/\delta)})$ and $\delta > 0$ is constant, and also $(\alpha, \varepsilon)$-Rényi-DP with $\varepsilon = O(\alpha n)$ for $\alpha > 0$ constant. These techniques have the additional advantage of extending easily to the continuous time case. We then focus our analysis on two classical MCMC algorithms based on the Langevin diffusion: the unadjusted Langevin algorithm and its stochastic gradient variant (see Section 5). In doing so, we address the open problem of obtaining uniform-in-time DP guarantees in a non-convex setting.

## 2. Preliminaries

In Bayesian statistics, the primary object of interest is the posterior distribution of the parameter $\theta \in E$ given an observed dataset $\mathcal{D} \in \mathcal{S}$. The posterior distribution, denoted as $\pi_{\mathcal{D}}$, is of the form

$$\pi_{\mathcal{D}}(B) = \int_B L_{\mathcal{D}}(\theta) \lambda(d\theta) \Big/ Z_{\mathcal{D}} \ , \quad B \in \mathcal{B}(E) \ ,$$

where $(E, \mathcal{B}(E))$ is a Borel space, $L_{\mathcal{D}}(\theta)$ is the likelihood associated with $\mathcal{D}$, $\lambda$ is a prior distribution for the parameter, and $Z_{\mathcal{D}} = \int_E L_{\mathcal{D}}(\theta) \eta(d\theta)$. In order to make practical use of the Bayesian framework, it is then essential to have access to key statistics of the posterior distribution, such as its moments. MCMC algorithms aim to solve this task relying on a Markov chain, $(X_n^{\mathcal{D}})_{n \geqslant 1}$, which has law that converges to $\pi_{\mathcal{D}}$ asymptotically in the number of iterations $n$. Throughout the paper, we shall denote the transition kernel of the Markov chain as $P_{\mathcal{D}} : E \times \mathcal{B}(E) \to [0, 1]$, where this means $X_n^{\mathcal{D}} \sim P_{\mathcal{D}}(X_{n-1}^{\mathcal{D}}, \cdot)$ and also that $X_n^{\mathcal{D}} \sim P_{\mathcal{D}}^n(X_0^{\mathcal{D}}, \cdot)$, where $X_0^{\mathcal{D}}$ is the initial state of the chain. In particular, MCMC algorithms can be used to estimate expectations $\pi_{\mathcal{D}}(f) := \int_E f(\theta) \pi_{\mathcal{D}}(d\theta)$, for some statistics $f : E \to \mathbb{R}$ thanks to associated Monte Carlo averages $\frac{1}{N} \sum_{n=1}^N f(X_n^{\mathcal{D}})$, in the sense that $\lim_{N \to \infty} \frac{1}{N} \sum_{n=1}^N f(X_n^{\mathcal{D}}) = \pi_{\mathcal{D}}(f)$ (Robert & Casella, 2004). There can be several possible outputs of an MCMC algorithm, such as $X_n^{\mathcal{D}}$, the state of the chain at time $n$, or $(X_1^{\mathcal{D}}, \ldots, X_n^{\mathcal{D}})$, the entire path up to time $n$, or also the Monte Carlo estimator $\frac{1}{N} \sum_{n=1}^N f(X_n^{\mathcal{D}})$. In either case, we can think of our MCMC methods as a randomised algorithm $\mathcal{A}(\mathcal{D})$, that is a random function of the dataset. We refer to Roberts & Rosenthal (2004) for an introduction to the main concepts required to obtain (asymptotically) valid MCMC algorithms.

We call *randomised algorithm* any function $\mathcal{D} \mapsto \mathcal{A}(\mathcal{D})$, such that $\mathcal{A}(\mathcal{D})$ is a random variable on a probability space $(\Omega, \mathbb{P}, \mathcal{F})$. We denote the law of a randomised algorithm $\mathcal{A}$ as $P_{\mathcal{A}(\mathcal{D})}$, that is defined for any measurable set $B$ as

$$P_{\mathcal{A}(\mathcal{D})}(B) := \mathbb{P}(\mathcal{A}(\mathcal{D}) \in B).$$

The framework of differential privacy (Dwork, 2006) compares the output of a randomised algorithm obtained giving two adjacent datasets as input. Various notions of adjacency between datasets can be introduced, where the most popular considers adjacent any two datasets that differ in only one entry. Below we state the definition of differential privacy.

**Definition 2.1.** A randomised algorithm $\mathcal{A}$ is $(\varepsilon, \delta)$-differentially private, for $\varepsilon, \delta \geqslant 0$, if for any measurable set $B \in \mathcal{B}(E)$

$$P_{\mathcal{A}(\mathcal{D})}(B) \leqslant e^\varepsilon P_{\mathcal{A}(\mathcal{D}')}(B) + \delta,$$

for any pair of adjacent datasets $\mathcal{D}, \mathcal{D}' \in \mathcal{S}$.

Following Definition 2.1, we say that a posterior distribution is $(\varepsilon, \delta)$-differentially private when, for all measurable sets $B$, it holds that

$$\pi_{\mathcal{D}}(B) \leqslant e^\varepsilon \pi_{\mathcal{D}'}(B) + \delta, \tag{1}$$

for all adjacent datasets $\mathcal{D}, \mathcal{D}'$. This is interpreted as a privacy guarantee of an i.i.d. sample from the posterior.

A related notion of privacy that we consider in this article is that of *Rényi differential privacy* (Mironov, 2017). In order to introduce the Rényi-DP we first need to define the Rényi divergence. For $\alpha > 1$ and two probability distributions, $P$ and $Q$, for which their Radon-Nikodym derivative is well defined, their Rényi divergence is

$$\mathsf{D}_\alpha(P\|Q) := \frac{1}{\alpha - 1} \log\left( \int_\Omega \left( \frac{\mathrm{d}P}{\mathrm{d}Q} \right)^\alpha dQ \right). \quad (2)$$

We then have the following definition.

**Definition 2.2.** A randomised algorithm $\mathcal{A}$ is $(\alpha, \varepsilon)$-Rényi differentially private , $\varepsilon \geqslant 0$ and $\alpha > 1$, if for all adjacent sets $\mathcal{D}, \mathcal{D}'$ it holds that $\mathsf{D}_\alpha(P_{\mathcal{A}(\mathcal{D})}\|P_{\mathcal{A}(\mathcal{D}')}) \leqslant \varepsilon$.

When $\alpha \to \infty$ we recover $(\varepsilon, 0)$-DP. Notice also that a $(\alpha, \varepsilon)$-Rényi-differentially private algorithm is also $(\varepsilon - \frac{1}{\alpha-1} \log \delta, \delta)$-differentially private for any $\delta \in (0,1)$ (see Proposition 3 in Mironov (2017)). Similarly to (1), we say a posterior distribution is $(\alpha, \varepsilon)$-Rényi-differentially private when $\mathsf{D}_\alpha(\pi_{\mathcal{D}}\|\pi_{\mathcal{D}'}) \leqslant \varepsilon$ for all adjacent datasets $\mathcal{D}, \mathcal{D}'$.

## 3. From convergence to differential privacy

This section establishes the connection between the DP of an MCMC algorithm and its convergence to the posterior distribution.

### 3.1. Differential privacy of a Markov chain after $n$ steps

Here we are interested in the following question: *what is the relation between the differential privacy of the $n$-th state of an MCMC algorithm, the differential privacy of the target distribution, and the total variation distance between them?*

In order to address the question above, we consider the randomised algorithm $\mathcal{A}_s(\mathcal{D}) = X_n^{\mathcal{D}}$, where $(X_k^{\mathcal{D}})_{k \in \mathbb{N}}$ is a Markov chain with transition kernel $P_{\mathcal{D}}$ and that is initialised from a probability distribution $\nu_{\mathcal{D}}$. The law of $X_n^{\mathcal{D}}$ is denoted as $\nu_{\mathcal{D}} P_{\mathcal{D}}^n(\cdot) := \int \nu_{\mathcal{D}}(\mathrm{d}x) P_{\mathcal{D}}^n(x, \cdot)$. If $\|\cdot\|_{\mathrm{TV}}$ denotes the total variation distance (see Definition A.1), for two families of probability distributions $\mu_{\mathcal{D}}$, $\nu_{\mathcal{D}}$ that satisfy $\|\mu_{\mathcal{D}} - \nu_{\mathcal{D}}\|_{\mathrm{TV}} \leqslant \beta$, we have that $(\varepsilon, \delta)$-DP of $\mu_{\mathcal{D}}$ implies $(\varepsilon, \delta + \beta(e^\varepsilon + 1))$-DP of $\nu_{\mathcal{D}}$ (see Proposition A.2 in Appendix A.1 for the proof of this result). In the case of MCMC algorithms, the law of $\mathcal{A}(\mathcal{D})$ depends on the number of iterations $n$, and so will its total variation distance to $\pi_{\mathcal{D}}$. Nevertheless, also in our case we can apply Proposition A.2 to address the question above. In this sense we make the following assumption, which requires a bound on the total variation distance that is uniform over datasets and decreasing in the number of iterations.

**Assumption 3.1** (Data-uniform convergence of the Markov chain). Consider a family of transition kernels $\{P_{\mathcal{D}} : \mathrm{E} \times \mathcal{B}(\mathrm{E}) \to [0,1] : \mathcal{D} \in \mathcal{S}\}$ and a family of initial distributions $\{\nu_{\mathcal{D}} : \mathcal{D} \in \mathcal{S}\}$. There exist a positive, decreasing function R such that $\lim_{m \to \infty} \mathsf{R}(m) = \underline{r} \geqslant 0$, and a constant $\zeta < \infty$ such that for all $\mathcal{D} \in \mathcal{S}$ and all $n \in \mathbb{N}$

$$\|\nu_{\mathcal{D}} P_{\mathcal{D}}^n - \pi_{\mathcal{D}}\|_{\mathrm{TV}} \leqslant \zeta \mathsf{R}(n) . \quad (3)$$

In particular, Assumption 3.1 requires that $\zeta$ and R are independent of the dataset. The function R is typically such that either $\lim_{m \to \infty} \mathsf{R}(m) = \underline{r} = 0$, that is the when case the MCMC algorithm is asymptotically exact, or $\underline{r} > 0$, that is the case when the MCMC algorithm is biased, for instance as a result of using an unadjusted discretisation scheme of a continuous time process. Under this assumption, we can obtain the following relations between the DP guarantees of $\mathcal{A}_s(\mathcal{D})$ and $\pi_{\mathcal{D}}$.

**Proposition 3.2.** *Consider the randomised algorithm $\mathcal{A}_s(\mathcal{D}) \sim \nu_{\mathcal{D}} P_{\mathcal{D}}^n$ and suppose Assumption 3.1 is verified. The following statements hold:*

*(i) If $\pi$ is $(\varepsilon, \delta)$-differentially private, then $\mathcal{A}_s(\mathcal{D})$ is $(\varepsilon, \delta + \kappa \mathsf{R}(n))$-differentially private with $\kappa = \zeta(e^\varepsilon + 1)$.*

*(ii) If $\mathcal{A}_s(\mathcal{D})$ is $(\varepsilon, \delta)$-differentially private for any $n$, then $\pi$ is $(\varepsilon, \delta + (1 + e^\varepsilon)\zeta \underline{r})$-differentially private.*

*Proof.* The first statement is obtained applying Proposition A.2. The second statement follows applying Proposition A.2 to obtain that $\pi_{\mathcal{D}}$ is $(\varepsilon, \delta + (1 + e^\varepsilon)\zeta \mathsf{R}(n))$-differentially private, then taking the limit as $n \to \infty$. $\qquad\square$

The first statement is similar to Proposition 3 in Wang et al. (2015). We note also that the second statement in Proposition 3.2 is still valid when the convergence bound (3) is not data-uniform.

A simple corollary of Proposition 3.2-(ii) is that any MCMC algorithm that is asymptotically exact, i.e. such that $\underline{r} = 0$, will fail to be $(\varepsilon, \delta)$-differentially private for some number of iterations $n$ when the posterior itself is not $(\varepsilon, \delta)$-differentially private. The following proposition gives a quantitative result in this direction.

**Proposition 3.3.** *Suppose Assumption 3.1 is verified. Let $\varepsilon \geqslant 0$, $\delta \in [0,1)$, and $\delta' \in (\delta, 1)$ and suppose $\pi_{\mathcal{D}}$ is not $(\varepsilon, \delta')$-differentially private. Let*

$$n^* = \inf\{n \in \mathbb{N} : (1 + e^\varepsilon)\zeta \mathsf{R}(n) \leqslant \delta' - \delta\}.$$

*Then, $\mathcal{A}_s(\mathcal{D}) \sim \nu_{\mathcal{D}} P_{\mathcal{D}}^n$ is $\underline{not}$ $(\varepsilon, \delta)$-differentially private for all $n \geqslant n^*$.*

*Proof.* The result follows applying Proposition A.3, that can be found in Appendix A.2. $\qquad\square$

Proposition 3.3 gives that $\mathcal{A}_s(\mathcal{D}) \sim \nu_\mathcal{D} P_\mathcal{D}^n$ is not $(\varepsilon, \delta)$-differentially private whenever it is sufficiently close to $\pi_\mathcal{D}$, assuming $\pi_\mathcal{D}$ is not $(\varepsilon, \delta')$-differentially private for some $\delta' > \delta$.

Finally, we show that $\nu_\mathcal{D} P_\mathcal{D}^n$ can be far from $\pi_\mathcal{D}$ when $\nu_\mathcal{D} P_\mathcal{D}^n$ is $(\varepsilon, \delta_n)$-differentially private, while the posterior distribution violates a weaker DP guarantee. In particular, we say that the posterior distribution is not $(\varepsilon, \delta)$-differentially private when there exist adjacent datasets $\mathcal{D}, \mathcal{D}' \in \mathcal{S}$ and a measurable set B such that $\pi_\mathcal{D}(\mathsf{B}) > e^\varepsilon \pi_{\mathcal{D}'}(\mathsf{B}) + \delta$.

**Proposition 3.4.** *Consider the randomised algorithm $\mathcal{A}(\mathcal{D}) \sim \nu_\mathcal{D} P_\mathcal{D}^n$ and assume it is $(\varepsilon, \bar{\delta})$-differentially private. Let $\{\pi_\mathcal{D} : \mathcal{D} \in \mathcal{S}\}$ be a family of posterior distributions that is not $(\varepsilon, \delta)$-differentially private for some $\delta > \bar{\delta}$. Then, there exists a dataset $\mathcal{D} \in \mathcal{S}$ such that*

$$\|\nu_\mathcal{D} P_\mathcal{D}^n - \pi_\mathcal{D}\|_{\mathrm{TV}} > \frac{e^{-\varepsilon}}{1 + e^{-\varepsilon}} (\delta - \bar{\delta}) . \qquad (4)$$

*Proof.* The statement follows by an application of Proposition A.4 in Appendix A.3. □

The statement of Proposition A.4 in Appendix A.3 gives more insight than the statement above. Indeed, it shows that, for any pair of adjacent datasets $\mathcal{D}, \mathcal{D}'$ for which there exists a measurable set B such that $\pi_\mathcal{D}(\mathsf{B}) > e^\varepsilon \pi_{\mathcal{D}'}(\mathsf{B}) + \delta$, (4) holds for either $\mathcal{D}$ or $\mathcal{D}'$. This means that the $n$-th state of the MCMC algorithm can be far from the posterior distribution for numerous datasets. In practice, a posterior distribution $\pi$ can violate $(\varepsilon, \delta)$-DP even for large $\varepsilon$ and $\delta$ unless it is carefully designed (Dimitrakakis et al., 2017). Hence, Proposition 3.4 shows that it is essential to design $\pi$ carefully.

### 3.2. Differential privacy of Monte Carlo estimators

We are now concerned with the task of releasing an estimate of an expectation $\pi_\mathcal{D}(f) := \int_\mathrm{E} f(x) \pi_\mathcal{D}(\mathrm{d}x)$ obtained running an MCMC algorithm for $N$ iterations. Specifically, we are given an observable $f : \mathrm{E} \to \mathbb{R}$ and we simulate a Markov chain $X^\mathcal{D}$ with transition kernel $P_\mathcal{D}$ to obtain the ergodic average $\frac{1}{N} \sum_{n=1}^N f(X_n^\mathcal{D})$. Similarly to the previous section, our strategy to obtain a DP guarantee will be based on an assumption that requires the ergodic average to be close to the truth, i.e. $\pi_\mathcal{D}(f)$.

We consider the randomised algorithm

$$\mathcal{A}(\mathcal{D}) = \frac{1}{N} \sum_{n=1}^N f(X_n^\mathcal{D}) + L, \qquad (5)$$

where $L$ is a random variable that is independent of the chain $X^\mathcal{D}$ and $f$ is the observable of interest. Here, it is crucial to add noise to the ergodic average to prevent an adversary from distinguishing between two adjacent datasets based on the observed output. In particular, $L$ should satisfy the following assumption, which makes the release of scalars that are close to each other differentially private.

**Assumption 3.5.** Let $\eta \in (0, \infty)$. There exist $(\varepsilon, \delta) \in \mathbb{R}_+ \times [0, 1)$ such that, for any measurable set B and any $a, b \in \mathbb{R}$ for which $|a - b| \leqslant \eta$, it holds

$$\mathbb{P}(a + L \in \mathsf{B}) \leqslant e^\varepsilon \, \mathbb{P}(b + L \in \mathsf{B}) + \delta.$$

A typical choice is to draw $L$ from the Laplacian distribution with scale parameter $\eta/\varepsilon$, which for any $\eta \in (0, \infty)$ satisfies Assumption 3.5 with parameters $(\varepsilon, 0)$ (Dwork et al., 2006).

In order to take advantage of Assumption 3.5, we shall require that the ergodic averages for the observable $f$ corresponding to two Markov chains for two adjacent datasets $\mathcal{D}, \mathcal{D}'$ are $\eta$-close with high probability. Without such a property, an adversary would be able to distinguish between two adjacent datasets based on the observed Monte Carlo estimator. We formalise this requirement in the next assumption.

**Assumption 3.6.** Let $\{P_\mathcal{D}^n : \mathcal{D} \in \mathcal{S}\}$ be a family of Markov kernels and $\{\nu_\mathcal{D} : \mathcal{D} \in \mathcal{S}\}$ a family of initial distributions and let $N \in \mathbb{N}$. There exist $\eta < \infty$, $\tilde{\delta} \in (0, 1)$ that are independent of $\mathcal{D}, \mathcal{D}'$, such that for any adjacent datasets $\mathcal{D}$ and $\mathcal{D}'$

$$\mathbb{P}\left(\left| \frac{1}{N} \sum_{n=1}^N f(X_n^\mathcal{D}) - \frac{1}{N} \sum_{n=1}^N f(X_n^{\mathcal{D}'}) \right| \leqslant \eta\right) \geqslant 1 - \tilde{\delta} ,$$

for some joint processes $(X_n^\mathcal{D}, X_n^{\mathcal{D}'})_{n=1}^N$ such that $(X_n^\mathcal{D})_{n=1}^N$ and $(X_n^{\mathcal{D}'})_{n=1}^N$ are two Markov chains respectively with transition kernels $P_\mathcal{D}$ and $P_{\mathcal{D}'}$, and initial distributions $\nu_\mathcal{D}, \nu_{\mathcal{D}'}$.

Note that in Assumption 3.6, we do not suppose that $X_n^\mathcal{D}$ and $X_n^{\mathcal{D}'}$ are independent and we can consider any coupling between $\nu_\mathcal{D} P_\mathcal{D}^n$ and $\nu_{\mathcal{D}'} P_{\mathcal{D}'}^n$. Moreover, we remark that Assumption 3.6 should hold only for the observable of interest.

In order to obtain a guarantee on the DP of $\mathcal{A}$, we shall assume that $L$ satisfies Assumption 3.5 with $\eta$ as given by Assumption 3.6. Notice that the variance of the noise injected in the output to achieve a fixed level of DP $\varepsilon$ increases with $\eta$. In particular, in the case of Laplacian noise we find

$$\mathrm{Var}(\mathcal{A}(\mathcal{D})) = \mathrm{Var}\left( \frac{1}{N} \sum_{n=1}^N f(X_n^\mathcal{D}) \right) + \frac{\eta}{\varepsilon} .$$

Therefore, the variance of the injected noise scales linearly with $\eta$.

We are now ready to state our result, that is obtained applying a more general argument that holds for any algorithm of the type $\mathcal{A}(\mathcal{D}) = g_\mathcal{D} + L$, where $g_\mathcal{D}$ is a random function that depends on $\mathcal{D}$ (see Proposition A.6 in Appendix A.4).

**Proposition 3.7.** *Let $f : E \to \mathbb{R}$ and let $\mathcal{A}$ be the corresponding randomised mechanism defined in* (5). *Suppose Assumption 3.6 holds for some $\eta \in (0, \infty)$, $\tilde{\delta} \in [0, 1)$, and that $L$ satisfies Assumption 3.5 with the same $\eta$ and some $\varepsilon \in \mathbb{R}_+$, $\delta \in [0, 1 - \tilde{\delta})$. Then, $\mathcal{A}$ is $(\varepsilon, \delta + \tilde{\delta})$-differentially private.*

*Proof.* The result follows applying Proposition A.6. $\square$

Verifying Assumption 3.6 directly would require ad-hoc arguments. Therefore we now introduce two conditions that are sufficient to ensure that Assumption 3.6 holds. We start with a high-probability, non-asymptotic bound on the convergence of the ergodic average to the true value, $\pi_{\mathcal{D}}(f)$.

**Assumption 3.8.** Let $f : E \to \mathbb{R}$. Let $\{P_{\mathcal{D}}^n : \mathcal{D} \in \mathcal{S}\}$ be a family of Markov kernels and $\{\nu_{\mathcal{D}} : \mathcal{D} \in \mathcal{S}\}$ a family of initial distributions and let $N \in \mathbb{N}$. Denote by $(X_n^{\mathcal{D}})_{n=1}^N$ a Markov chain with transition kernel $P_{\mathcal{D}}$ for any $\mathcal{D} \in \mathcal{S}$. There exist $\tilde{\delta} \in (0, 1)$ and $C < \infty$ such that for any $\mathcal{D} \in \mathcal{S}$

$$\mathbb{P}\left( \left| \frac{1}{N} \sum_{n=1}^N f(X_n^{\mathcal{D}}) - \pi_{\mathcal{D}}(f) \right| \leqslant C \right) \geqslant 1 - \tilde{\delta}.$$

This assumption can be typically verified using some mixing properties of the family of Markov kernels $\{P_{\mathcal{D}}^n : \mathcal{D} \in \mathcal{S}\}$, that should be uniform over $\mathcal{D} \in \mathcal{S}$ as stated in Assumption 3.1; see e.g. Paulin (2015); Durmus et al. (2023) and the reference therein.

Then, we require that the absolute value of the difference between $\pi_{\mathcal{D}}(f)$ and $\pi_{\mathcal{D}'}(f)$ is bounded by a constant for any adjacent datasets $\mathcal{D}, \mathcal{D}'$. This ensures that the output $\mathcal{A}$ localises around similar values and thus can be private.

**Assumption 3.9.** There exists $\gamma_f < \infty$ such that

$$|\pi_{\mathcal{D}}(f) - \pi_{\mathcal{D}'}(f)| \leqslant \gamma_f,$$

for any adjacent datasets $\mathcal{D}, \mathcal{D}' \in \mathcal{S}$.

We now state our main result of the section.

**Proposition 3.10.** *Suppose Assumptions 3.8 and 3.9 hold for some $C, \tilde{\delta}, \gamma_f$. Suppose also that $L$ satisfies Assumption 3.5 for $\eta = 2C + \gamma_f$. Then the randomised mechanism $\mathcal{A}$ defined in* (5) *is $(\varepsilon, \delta + 2\tilde{\delta} - \tilde{\delta}^2)$-differentially private.*

*Proof.* The proof is based on showing that Assumption 3.6 holds under our assumptions. The details can be found in Appendix A.5. $\square$

# 4. Characterising the Privacy of Diffusions

In this section we describe our proof strategy to obtain DP guarantees of MCMC algorithms based on diffusions. We focus here on Radon-Nikodym derivatives, which is essentially an abstraction of the well-known approach using the density function of the random mechanism, see for example in Section 3.1 of Desfontaines & Pejó (2022) or Lemma 7.1.5 of Vadhan (2017) for Lemma 4.1. This framework allows us to calculate privacy parameters via Girsanov's theorem, which describes a change of measure on the probability space.

## 4.1. Differential Privacy with Radon-Nikodym Derivatives

We shall start describing our approach considering an abstract randomised algorithm $\mathcal{A}(\mathcal{D}) : \Omega \to \mathcal{X}$ taking values in some (measurable) space $\mathcal{X}$ and defined on a probability space $(\Omega, \mathcal{F}, \mathbb{P})$. This shall be interpreted in the sequel as the output of an MCMC algorithm such as its $n$-th state or its full path up to the the $n$-th state. Here we stress the dependence on $\mathbb{P}$ and denote the distribution of $\mathcal{A}(\mathcal{D})$ for any $\mathcal{D} \in \mathcal{S}$ as

$$P_{\mathcal{A}(\mathcal{D})}^{\mathbb{P}}(\mathsf{B}) = \mathbb{P}(\mathcal{A}(\mathcal{D}) \in \mathsf{B}),$$

for any measurable set $\mathsf{B} \subset \mathcal{X}$. Our strategy to obtain DP of the mechanism $\mathcal{A}$ is based on a change of measure argument. In particular, for every pair of adjacent datasets $\mathcal{D}, \mathcal{D}' \in \mathcal{S}$ we shall find a probability measure $\mathbb{Q}$ on $(\Omega, \mathcal{F})$ such that the law of $\mathcal{A}(\mathcal{D})$ under $\mathbb{Q}$ is equal to the law of $\mathcal{A}(\mathcal{D}')$ under $\mathbb{P}$. This is to say,

$$P_{\mathcal{A}(\mathcal{D})}^{\mathbb{P}} = P_{\mathcal{A}(\mathcal{D}')}^{\mathbb{Q}}. \tag{6}$$

Note that $\mathbb{Q}$ depends on $\mathcal{D}, \mathcal{D}'$, but to avoid overloading the notation, we keep this dependence implicit. When $\mathbb{P}$ and $\mathbb{Q}$ are mutually absolutely continuous, we can define the Radon-Nikodym (RN) derivative $\frac{d\mathbb{Q}}{d\mathbb{P}} : \Omega \to \mathbb{R}$, together with its inverse $\frac{d\mathbb{P}}{d\mathbb{Q}}$. That is to say, for every random variable $Z : \Omega \to \mathcal{X}$ one has

$$\mathbb{E}_{\mathbb{Q}}[Z] = \mathbb{E}\left[ Z \frac{d\mathbb{Q}}{d\mathbb{P}} \right],$$

where $\mathbb{E}_{\mathbb{Q}}$ denotes integration on $\Omega$ with respect to $\mathbb{Q}$, while $\mathbb{E}$ denotes integration with respect to $\mathbb{P}$. Since $\frac{d\mathbb{Q}}{d\mathbb{P}}$ is a mapping from the probability space, we can treat it as a random variable taking values in $\mathbb{R}$.

Now that we have set the framework, we can show how to obtain DP of $\mathcal{A}$ bounding the Radon-Nikodym derivative with high probability.

**Lemma 4.1.** *Let $\mathcal{A}$ be a random mechanism and suppose for every two adjacent datasets there exists a measure $\mathbb{Q}$ such that* (6) *holds. Suppose furthermore for every such $\mathbb{Q}$ the Radon-Nikodym derivative, $\frac{d\mathbb{P}}{d\mathbb{Q}}$, is well defined, and also that*

$$\mathbb{P}\left( \frac{d\mathbb{P}}{d\mathbb{Q}} > e^\varepsilon \right) \leqslant \delta.$$

*Then, $\mathcal{A}$ is $(\varepsilon, \delta)$-differentially private.*

*Proof.* The proof follows by splitting the expectation of the relevant indicator function into regions where $\frac{d\mathbb{P}}{d\mathbb{Q}}$ is large and small. Full details can be found in Appendix B.1. $\square$

In the next section, we will rely on Lemma 4.1 to obtain DP guarantees for diffusion-based MCMC algorithms, where the Radon-Nikodym derivative can be obtained by Girsanov's theorem. The following lemma is essentially a corollary of the data processing inequality, see Theorem 9 in van Erven & Harremos (2014).

**Lemma 4.2.** *Let $\mathcal{A}$ be a random mechanism and consider two constants $\alpha > 1, \varepsilon > 0$. Suppose for every $\mathcal{D}, \mathcal{D}' \in \mathcal{S}$ there exists a measure $\mathbb{Q}$ such that (6) holds. Suppose in addition that the Radon-Nikodym derivative $\frac{d\mathbb{P}}{d\mathbb{Q}}$ is well defined and*

$$\mathbb{E}\left[\left(\frac{d\mathbb{P}}{d\mathbb{Q}}\right)^{\alpha-1}\right] \leqslant e^{(\alpha-1)\varepsilon}.$$

*Then, $\mathcal{A}$ is $(\alpha, \varepsilon)$-Rényi differentially private.*

*Proof.* The proof can be found in Appendix B.2. $\square$

*Remark* 4.3. For all results in this paper, bounds in $(\varepsilon, \delta)$-DP can be achieved by converting a Rényi-DP bound, as mentioned in Section 2. However, we presented Lemmas 4.1 and 4.2 separately since Lemma 4.2 applies more generally, and in particular does not require control of the tails of $d\mathbb{P}/d\mathbb{Q}$.

## 4.2. Differential privacy of stochastic differential equations

In this section we shall consider the DP of random mechanisms $\mathcal{A}$ that involve the solution of a stochastic differential equation (SDE). We are particularly interested in two distinct random mechanisms generated by (7): the single time evaluation $\mathcal{A}_s(\mathcal{D}) = X_T^{\mathcal{D}}$ taking values in $\mathbb{R}^d$, and also the whole path $\mathcal{A}_p(\mathcal{D}) = (X_t^{\mathcal{D}})_{t \in [0,T]}$ taking values in $C([0,T], \mathbb{R}^d)$, that is the space of continuous functions on $[0, T]$ taking values in $\mathbb{R}^d$.

The general SDE we study is

$$dX_t^{\mathcal{D}} = f_{\mathcal{D}}(X_{\kappa(t)}^{\mathcal{D}}, \eta_{\kappa(t)})dt + \sqrt{2/\beta}\, dW_t, \qquad (7)$$

with initial condition $X_0^{\mathcal{D}} = x_0 \in \mathbb{R}^d$. Here $W_t$ is Brownian motion, $f_{\mathcal{D}} : \mathbb{R}^d \times \mathcal{Y} \to \mathbb{R}^d$ is a measurable function, $\kappa : [0, \infty) \to [0, \infty)$ is a function satisfying $\kappa(t) \leqslant t$, $(\eta_t)_{t \geqslant 0}$ is a stochastic process independent of $(W_t)_{t \geqslant 0}$ taking values in some space $\mathcal{Y}$, and $\beta > 0$ is an inverse temperature parameter that scales the noise. The function $\kappa$ allows us to consider discrete time approximations obtained

e.g. applying the Euler scheme to a continuous time SDE. For instance, the choice $\kappa(t) := \gamma\lfloor t/\gamma \rfloor$ corresponds to the backwards projection onto the grid $\{n\gamma\}_{n \in \mathbb{N}}$. In this case one has that $X_{n\gamma}^{\mathcal{D}}$ is equal in law to the Markov chain $(x_n^{\mathcal{D}})_{n \in \mathbb{N}}$

$$x_{n+1}^{\mathcal{D}} = x_n^{\mathcal{D}} + \gamma f_{\mathcal{D}}(x_n^{\mathcal{D}}, \eta_{n\gamma}) + \sqrt{2\gamma/\beta}\, z_{n+1},$$

and $(z_n)_{n \geqslant 1}$ is a sequence of independent standard normal random variables. The process $(\eta_t)_{t \geqslant 0}$ allows us to consider additional sources of randomness, for instance the random mini-batches that give the stochastic gradient in the noisy SGD algorithm.

*Remark* 4.4. We assume without further commentary that (7) has a unique strong solution. This holds when $\kappa$ is a projection onto a grid, and also when $\kappa(t) = t$ under weak conditions (Zhang, 2005).

In the following result, we demonstrate that our proof strategy can be used to give privacy bounds for $\mathcal{A}_s(\mathcal{D})$ that are uniform in $T > 0$. We remark that the condition (9) can be satisfied in non-convex settings, which to the best of our knowledge have not been addressed thus far in the literature.

**Proposition 4.5** (Privacy of the final value). *Let $T > 0$ and assume there exist constants $L, C, c > 0$ such that for every adjacent datasets $\mathcal{D}, \mathcal{D}' \in \mathcal{S}$ one has for any $x, y \in \mathbb{R}^d$ and $s \in \mathcal{Y}$*

$$|f_{\mathcal{D}}(x, s) - f_{\mathcal{D}'}(y, s)| \leqslant L|x - y| + c, \qquad (8)$$

*and $(X_t^{\mathcal{D}})_{t \in [0,T]}$ and $(X_t^{\mathcal{D}'})_{t \in [0,T]}$ are solutions of (7) with drifts $f_{\mathcal{D}}$ and $f_{\mathcal{D}'}$ respectively, driven by the same Brownian motion $(W_t)_t \geqslant 0$, such that almost surely[1] it holds that*

$$\sup_{t \in [0,T]} |X_t^{\mathcal{D}} - X_t^{\mathcal{D}'}| \leqslant C. \qquad (9)$$

*Then for $\delta > 0, \alpha \geqslant 1$ one has that $\mathcal{A}_s(\mathcal{D}) = X_T^{\mathcal{D}}$ is $(\varepsilon_\delta, \delta)$-private and $(\alpha, \varepsilon_\alpha)$-Rényi private for*

$$\varepsilon_\delta = C_2/4 + \sqrt{C_2 \log(1/\delta)}, \quad \varepsilon_\alpha = \alpha C_2/4,$$

*where $C_2 = \beta(C(L + 1) + c)^2$.*

*Proof.* We fix a pair of adjacent datasets $\mathcal{D}, \mathcal{D}' \in \mathcal{S}$ and $T > 0$, and define an auxiliary process $(Z_t)_{t \in [0,T]}$ satisfying

$$Z_t = X_t^{\mathcal{D}} \text{ for } t \in [0, T-1], \qquad Z_T = X_T^{\mathcal{D}'}.$$

In particular, $Z_t$ is given by perturbing the dynamics of $X_t^{\mathcal{D}}$ on $t \in [T-1, T]$ in such a way that it approaches $X_T^{\mathcal{D}'}$. By the assumption that $X_t^{\mathcal{D}}$ and $X_t^{\mathcal{D}'}$ are almost

---

[1] Since the measures $\mathbb{P}$ and $\mathbb{Q}$ are absolutely continuous, here and elsewhere we don't distinguish for which measure the event in question is almost sure.

surely close, one may define this perturbation in such a way that it is almost surely bounded. Therefore via Girsanov's theorem we can find a measure $\mathbb{Q}$ satisfying the assumptions of Lemmas 4.1 and 4.2. Full details are presented in Appendix B.6. $\qquad\square$

Furthermore, we have the following result on the privacy of the entire trajectory of $X^{\mathcal{D}}$ up to a time $T > 0$.

**Proposition 4.6** (Privacy of the path). *Consider the family of processes* (7). *Suppose there exists* $c > 0$ *such that for every adjacent datasets* $\mathcal{D}, \mathcal{D}' \in \mathcal{S}$ *and* $x \in \mathbb{R}^d, \eta \in \mathcal{Y}$ *one has*

$$|f_{\mathcal{D}}(x, \eta) - f_{\mathcal{D}'}(x, \eta)| \leqslant c . \tag{10}$$

*Then for* $\delta > 0, \alpha \geqslant 1$ *one has that the algorithm* $\mathcal{A}_p(\mathcal{D}) = (X_T^{\mathcal{D}})_{t \in [0,T]}$ *is* $(\varepsilon_\delta, \delta)$-*differentially private and* $(\alpha, \varepsilon_\alpha)$-*Rényi differentially private for*

$$\varepsilon_\delta = C_1(T)/4 + \sqrt{C_1(T) \log(1/\delta)}, \ \ \varepsilon_\alpha = \alpha C_1(T)/4 ,$$

*and* $C_1(T) = c^2 \beta T$.

*Proof.* Fixing adjacent datasets $\mathcal{D}, \mathcal{D}' \in \mathcal{S}$, we use Girsanov's theorem to define a measure $\mathbb{Q}$ under which $(X_t^{\mathcal{D}'})_{t \in [0,T]}$ is equal in distribution to $(X_t^{\mathcal{D}})_{t \in [0,T]}$ under $\mathbb{P}$. Using bounds on the assumed discrepancy between the drifts (10), we may control the Radon-Nikodym derivative $d\mathbb{P}/d\mathbb{Q}$ in such a way as to obtain the result applying Lemmas 4.1 and 4.2. Full details are presented in Appendix B.5. $\qquad\square$

# 5. Privacy of Langevin-based algorithms in the non-convex setting

In this section we obtain (Rényi)-DP for both the trajectory and the final value of two Langevin-based MCMC algorithms: ULA and noisy SGD. All bounds in this section are derived from Propositions 4.5 and 4.6.

## 5.1. Sampling from Bayesian posteriors with ULA

Consider a posterior distribution with density with respect to the Lebesgue measure on $\mathbb{R}^d$ of the form

$$\pi_{\mathcal{D}}(x) \propto e^{-U_{\mathcal{D}}(x)}, \tag{11}$$

for $U_{\mathcal{D}} : \mathbb{R}^d \to \mathbb{R}$. We can approximately sample from $\pi_{\mathcal{D}}$ with ULA, that is the Markov chain

$$x_{n+1}^{\mathcal{D}} = x_n^{\mathcal{D}} - \gamma \nabla U_{\mathcal{D}}(x_n^{\mathcal{D}}) + \sqrt{2\gamma}\, z_{n+1}, \tag{12}$$

with initial condition $x_0^{\mathcal{D}} = x_0 \in \mathbb{R}^d$, and where $(z_n)_{n \geqslant 1}$ is a sequence of i.i.d. standard Gaussians on $\mathbb{R}^d$, and $\gamma > 0$ is the step size. ULA arises as the Euler discretisation of the overdamped Langevin diffusion and has been shown to be successful for sampling from Bayesian posteriors under

a range of assumptions (Neal, 1992; Roberts & Tweedie, 1996; Durmus & Moulines, 2019). We shall consider the privacy of both the final draw $x_n^{\mathcal{D}} \in \mathbb{R}^d$, and of the entire chain $(x_1^{\mathcal{D}}, ..., x_n^{\mathcal{D}}) \in \mathbb{R}^{dn}$, under a non-convex assumption on $U_{\mathcal{D}}$. The particular assumption we place upon the posterior in the following theorem is close to the convexity outside of a ball condition considered in Durmus & Moulines (2017) and Erdogdu et al. (2022). The strongly convex part $K$ could be interpreted as a regulariser.

**Assumption 5.1.** Consider a posterior distribution (11) and suppose for every dataset $\mathcal{D} \in \mathcal{S}$ one may write $U_{\mathcal{D}} = V_{\mathcal{D}} + K$, where $V_{\mathcal{D}}, K : \mathbb{R}^d \to \mathbb{R}$ are continuously differentiable functions. In addition suppose that there exists a constant $c > 0$ such that $\sup_{x \in \mathbb{R}^d} |\nabla V_{\mathcal{D}}(x)| \leqslant c$. Furthermore, $\nabla K$ is $L$-Lipschitz, that is

$$|\nabla K(x) - \nabla K(y)| \leqslant L|x - y|,$$

for all $x, y \in \mathbb{R}^d$, and also strongly convex, that is there exists $a > 0$ such that for all $x, y \in \mathbb{R}^d$.

$$\langle \nabla K(x) - \nabla K(y), x - y \rangle \geqslant a\|x - y\|^2. \tag{13}$$

**Theorem 5.2.** *Suppose Assumption 5.1 holds and let* $\mathcal{A}_s(\mathcal{D}) = x_n^{\mathcal{D}}$ *be the* $n$-*th state of the ULA targeting* $\pi_{\mathcal{D}}$ *with step-size* $\gamma \in (0, 2a/L^2)$. *Then, for* $\delta > 0$ *and* $\alpha \geqslant 1$, $\mathcal{A}_s(\mathcal{D})$ *is* $(\varepsilon_\delta, \delta)$-*differentially private and* $(\alpha, \varepsilon_\alpha)$-*Rényi differentially private for*

$$\varepsilon_\delta = C_3/4 + \sqrt{C_3 \log(1/\delta)}, \ \ \varepsilon_\alpha = \alpha C_3/4,$$

*where* $C_3 = c^2 (\frac{2(L+1)}{a - \gamma L^2/2} + 1)^2$.

*Proof.* The proof here uses Proposition 4.5. One considers a continuous time interpolation of the ULA algorithm (12), at which point it just suffices to show that the closeness condition (9) holds for the continuous time interpolation. Full details are given in Appendix C.1. $\qquad\square$

Note that the above bound does *not* depend on the number of steps. Furthermore, since recent analysis like (Chewi et al., 2022) suggests that ULA recovers the true posterior in Rényi divergence as $\gamma \to 0$ and $n \to \infty$, taking the limit as $\gamma \to 0$ in Theorem 5.2 suggests that true posterior is differentially private with $C_3 > 0$ replaced with $C_4 = c(\frac{2(L+1)}{a} + 1)$.

Now we consider the privacy of the entire chain under slightly different assumptions.

**Assumption 5.3.** There exists a constant $c > 0$ such that the posterior $\pi_{\mathcal{D}}$ in (11) satisfies for any adjacent datasets $\mathcal{D}, \mathcal{D}' \in \mathcal{S}$ and all $x \in \mathbb{R}^d$

$$|\nabla U_{\mathcal{D}}(x) - \nabla U_{\mathcal{D}'}(x)| \leqslant c .$$

Note that the following result places no requirement on the step size, but it does depend on the number of steps.

**Theorem 5.4.** *Suppose Assumption 5.3 holds and let $\mathcal{A}_p(\mathcal{D}) = (x_1^{\mathcal{D}}, \ldots, x_n^{\mathcal{D}})$ be the path of ULA up to state $n \in \mathbb{N}$. Then, for $\delta > 0, \alpha \geqslant 1$, the algorithm $\mathcal{A}_p(\mathcal{D})$ is $(\varepsilon_\delta, \delta)$-differentially private and $(\alpha, \varepsilon_\alpha)$-Rényi differentially private for*

$$\varepsilon_\delta = C_5(n)/4 + \sqrt{C_5(n)\log(1/\delta)}, \;\; \varepsilon_\alpha = \alpha C_5(n)/4 ,$$

*where $C_5(n) = n\gamma c^2$.*

*Proof.* We use the continuous time interpolation of (12), along with Proposition 4.6. Then since $(x_1^{\mathcal{D}}, ..., x_n^{\mathcal{D}})$ is equal in law to a mapping from $(X_T^{\mathcal{D}})_{t \in [0, n\gamma]}$, the result follows from the data processing inequality (see Theorem 9 in van Erven & Harremos (2014)). $\qquad\square$

### 5.2. Noisy stochastic gradient descent

In this section we study the DP of a stochastic-gradient variant of the ULA. This algorithm is essentially a noisy version of the stochastic gradient descent and can be used to minimise the loss function

$$\mathcal{L}_{\mathcal{D}}(x) := \frac{1}{m}\sum_{i=1}^{m} \ell(x, d_i) , \tag{14}$$

where we assumed the dataset is of the form $\mathcal{D} = \{d_1, ..., d_m\}$. The algorithm we consider is driven by the following Markov chain:

$$x_{n+1}^{\mathcal{D}} = x_n^{\mathcal{D}} - \frac{\gamma}{s}\sum_{i \in A_{n+1}} \nabla_x \ell(x_n^{\mathcal{D}}, d_i) + \sqrt{2\gamma/\beta}z_{n+1} , \tag{15}$$

where $\ell : \mathbb{R}^d \times \mathcal{Y} \to \mathbb{R}$, $(A_n)_{n \geqslant 1}$ is a sequence of independent random variables uniformly distributed on subsets of $\{1, ..., m\}$ of size $s \leqslant m$, the step size is $\gamma > 0$, and $(z_n)_{n \geqslant 1}$ is a sequence of i.i.d. standard Gaussians on $\mathbb{R}^d$, independent of $(A_n)_{n \geqslant 1}$. Here $\ell(x, d_i)$ is interpreted as the loss incurred for datum $d_i$. We remark that the algorithm (15) is known in the MCMC literature as stochastic gradient Langevin dynamics (Welling & Teh, 2011).

In order to prove DP of the algorithm, we shall work under the following assumption on the loss function.

**Assumption 5.5.** Consider the loss function (14), where

$$\ell(x, d) = v(x, d) + k(x) ,$$

for functions $v : \mathbb{R}^d \times \mathcal{S} \to \mathbb{R}$ and $k : \mathbb{R}^d \to \mathbb{R}$ that are once continuously differentiable. Furthermore, $\nabla k$ is $L$-Lipschitz and for $c, a > 0$ one has $|\nabla_x v(x, d)| \leqslant c$ for any $x \in \mathbb{R}^d$ and any datum $d$, as well as

$$\langle \nabla k(x) - \nabla k(y), x - y \rangle \geqslant a\|x - y\|^2 .$$

We then have the following result.

**Theorem 5.6.** *Suppose Assumption 5.5 holds and consider the algorithm $\mathcal{A}_s(\mathcal{D}) = x_n^{\mathcal{D}}$ described in (15) with $\gamma \in (0, 2a/L^2)$ and stochastic gradient of size $s \leqslant m$. Then, for $\delta > 0$ and $\alpha \geqslant 1$, one has that $\mathcal{A}_s$ is $(\varepsilon_\delta, \delta)$-differentially private and $(\alpha, \varepsilon_\alpha)$-Rényi differentially private for*

$$\varepsilon_\delta = C_6/4 + \sqrt{C_6\log(1/\delta)}, \quad \varepsilon_\alpha = \alpha C_6/4 ,$$

*where $C_6 = c^2\beta\left(\frac{2(L+1)}{a - \gamma L^2/2} + 1\right)^2$.*

*Proof.* The proof here is similar to the proof of Theorem 5.2, with minor alterations due to the stochastic gradient and the inverse temperature parameter $\beta > 0$. Full details are given in Appendix C.2. $\qquad\square$

Theorem 5.6 is only a minor refinement of Theorem 5.2, and in particular the privacy guarantee does not improve with the size $m$ of the dataset. However, in the strongly convex case where each $\nabla v$ is constant, the following theorem (similar to Theorem 2 in Chourasia et al. (2021)) shows that privacy does increase with $s \leqslant m$.

**Theorem 5.7.** *Consider the setting of Theorem 5.6, but suppose that for any datum $d \in \mathcal{D}$ with $\mathcal{D} \in \mathcal{S}$ one has that $\nabla_x v(x, d)$ is constant in $x \in \mathbb{R}^d$. Then, for $\delta > 0$ and $\alpha \geqslant 1$, one has that $\mathcal{A}_s$ is $(\varepsilon_\delta, \delta)$-differentially private and $(\alpha, \varepsilon_\alpha)$-Rényi differentially private for*

$$\varepsilon_\delta = C_7 + \sqrt{C_7\log(1/\delta)}, \quad \varepsilon_\alpha = \alpha C_7/4 ,$$

*where $C_7 = \frac{c^2\beta}{s^2}\left(\frac{2(L+1)}{a - \gamma L^2/2} + 1\right)^2$.*

*Proof.* The proof can be found in Appendix C.3. $\qquad\square$

It is an open problem as to how $m$ and $s$ affect the privacy of the noisy SGD algorithm in the setting of Theorem 5.6. However, the following result on the path of noisy SGD improves with the size $s \leqslant m$ of the stochastic gradient.

**Theorem 5.8.** *Consider the loss function (14), and suppose there exists $c > 0$ such that for every $\mathcal{D}, \mathcal{D}' \in \mathcal{S}$ and $d \in \mathcal{D}, d' \in \mathcal{D}'$ one has that*

$$|\nabla_x \ell(x, d) - \nabla_x \ell(x, d')| \leqslant c.$$

*Then, for $\delta > 0$ and $\alpha \geqslant 1$, the algorithm $\mathcal{A}_p(\mathcal{D}) = (x_1^{\mathcal{D}}, \ldots, x_n^{\mathcal{D}})$ shown in (15) is $(\varepsilon_\delta, \delta)$-differentially private and $(\alpha, \varepsilon_\alpha)$-Rényi differentially private for*

$$\varepsilon_\delta = C_8 + \sqrt{C_8\log(1/\delta)}, \quad \varepsilon_\alpha = \alpha C_8/4 ,$$

*where $C_8(n) = \frac{\beta c^2}{s^2}n\gamma$.*

*Proof.* The proof is very similar to the proof of Theorem 5.4. Full details are given in Appendix C.4. $\qquad\square$

The results of Ryffel et al. (2022) and Ye & Shokri (2022) suggest that one may obtain superior bounds by fully exploiting the randomness of the stochastic gradient. However, for simplicity we do not consider this in the current work.

### 5.3. Commentary on Bounds Presented

We have studied two kinds of dimension-independent privacy guarantees: for the final draw, and for the entire trajectory of an MCMC chain. In particular, our bounds for the privacy of the final draw in Theorems 5.2 and 5.6 are uniform-in-time in a non-convex setting on an unbounded space, which addresses Question 1.2 and provides an upper bound for Question 1.1 in Altschuler & Talwar (2022). Also, unlike the results of Altschuler & Talwar (2022), our bounds do not depend on the size of the state space (which is unbounded), and do not blow up as the step-size goes to 0. In this sense, we therefore generalise the results of Chourasia et al. (2021) to a non-convex setting. Indeed, our results match Chourasia et al. (2021) in the strongly convex regime.

On the other hand, our results for the entire trajectory we improved on known composition bounds for $(\epsilon, \delta)$-privacy, matching composition bounds for Rényi privacy. This removes the need for complicated $(\epsilon, \delta)$-DP privacy accounting, and provides a simple framework which extends naturally to the continuous time case. In particular, our $(\varepsilon, \delta)$-DP bounds in Theorems 5.2 and 5.6 feature uniform values of $\delta > 0$, and $\varepsilon = O(n + \sqrt{n \log(1/\delta)})$ dependence on the number of steps $n$, whilst the advanced composition bounds presented in Kairouz et al. (2015) achieve $\varepsilon = O(n + \sqrt{n \log(e + (\varepsilon \sqrt{n}/\delta))})$.

## 6. Conclusions

We have presented a variety of results on the differential privacy of MCMC algorithms. Our results clarify the importance of choosing a Bayesian posterior distribution that has good DP guarantees, or else an MCMC algorithm cannot be expected to both private and close to convergence. Our results imply that it is crucial to design the MCMC algorithm together with the Bayesian model, in order to allow an end-to-end private inference. We have also discussed a novel approach to prove DP based on bounding the Radon-Nikodym derivative of the algorithm. This strategy allowed us to obtain new non-convex results that extend the known privacy properties of Langevin-based MCMC algorithm. We expect that a more careful analysis could allow extensions of our results to more general assumptions on the posterior distribution. Our approach can be applied to very general randomised algorithms and not only to those considered in this article. We leave extensions to more complex settings such as Bayesian federated learning for future work.

## Acknowledgements

All authors acknowledge funding by the European Union (ERC-2022-SyG, 101071601). Views and opinions expressed are however those of the authors only and do not necessarily reflect those of the European Union or the European Research Council Executive Agency. Neither the European Union nor the granting authority can be held responsible for them. Gareth O. Roberts has been supported by the UKRI grant EP/Y014650/1 as part of the ERC Synergy project OCEAN, EPSRC grants Bayes for Health (R018561), CoSInES (R034710), PINCODE (EP/X028119/1), ProbAI (EP/Y028783/1) and EP/V009478/1.

We thank Mengchu Li and Shenggang Hu for the useful discussions.

## Impact Statement

This paper presents work whose goal is to advance the field of Machine Learning. We believe by extending the theoretical study of differential privacy, practitioners will be able to design methods that better protect the privacy of individuals.

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

# A. Proofs and general results for Section 3

In this section we give proofs and general statements relative to Section 3. These rely on the following definition.

**Definition A.1.** Let $\mu$ and $\nu$ be two probability distributions on $(E, \mathcal{B}(E))$. Their total variation (TV) distance is

$$\|\mu - \nu\|_{\mathrm{TV}} := \sup_{B \in \mathcal{B}(E)} |\mu(B) - \nu(B)|.$$

## A.1. Auxiliary result for Proposition 3.2

The following result is analogous to Proposition 12 in Minami et al. (2016).

**Proposition A.2.** *Let $\{\mu_{\mathcal{D}} : \mathcal{D} \in \mathcal{S}\}$ and $\{\nu_{\mathcal{D}} : \mathcal{D} \in \mathcal{S}\}$ be two families of probability distributions that satisfy $\|\mu_{\mathcal{D}} - \nu_{\mathcal{D}}\|_{\mathrm{TV}} \leqslant \beta$ for any $\mathcal{D} \in \mathcal{S}$. If $\{\mu_{\mathcal{D}} : \mathcal{D} \in \mathcal{S}\}$ is $(\varepsilon, \delta)$-differentially private, then $\{\nu_{\mathcal{D}} : \mathcal{D} \in \mathcal{S}\}$ is $(\varepsilon, \delta + \beta(e^{\varepsilon} + 1))$-differentially private.*

*Proof.* Let $\mathcal{D}, \mathcal{D}' \in \mathcal{S}$ denote two adjacent datasets and $B$ be a measurable set. First, we observe that under our assumption and by the definition of the total variation distance we have $|\mu_{\mathcal{D}}(B) - \nu_{\mathcal{D}}(B)| \leqslant \beta$. This implies following inequalities for any $\mathcal{D}$:

$$\mu_{\mathcal{D}}(B) \leqslant \nu_{\mathcal{D}}(B) + \beta, \tag{16}$$
$$\nu_{\mathcal{D}}(B) \leqslant \mu_{\mathcal{D}}(B) + \beta. \tag{17}$$

Using these inequalities as well as the $(\varepsilon, \delta)$-DP of $\mu$, we find that

$$\begin{aligned} \nu_{\mathcal{D}}(B) &\leqslant \mu_{\mathcal{D}}(B) + \beta \\ &\leqslant e^{\varepsilon} \mu_{\mathcal{D}'}(B) + \delta + \beta \\ &\leqslant e^{\varepsilon} (\nu_{\mathcal{D}'}(B) + \beta) + \delta + \beta, \end{aligned}$$

which concludes the proof. $\qquad\square$

## A.2. General result for Proposition 3.3

**Proposition A.3.** *Let $\{\mu_{\mathcal{D}} : \mathcal{D} \in \mathcal{S}\}$ and $\{\nu_{\mathcal{D}} : \mathcal{D} \in \mathcal{S}\}$ be two families of probability distributions that satisfy $\|\mu_{\mathcal{D}} - \nu_{\mathcal{D}}\|_{\mathrm{TV}} \leqslant \beta$ for any $\mathcal{D} \in \mathcal{S}$. Let $\varepsilon > 0$, $\delta \in [(1 + e^{\varepsilon})\beta, 1)$, and suppose $\{\nu_{\mathcal{D}} : \mathcal{D} \in \mathcal{S}\}$ is not $(\varepsilon, \delta)$-differentially private. Then, $\{\mu_{\mathcal{D}} : \mathcal{D} \in \mathcal{S}\}$ is not $(\varepsilon, \delta - (1 + e^{\varepsilon})\beta)$-differentially private.*

*Proof.* Let $\mathcal{D}, \mathcal{D}' \in \mathcal{S}$ denote two adjacent datasets and $B$ be a measurable set such that

$$\nu_{\mathcal{D}'}(B) \geqslant e^{\varepsilon} \nu_{\mathcal{D}}(B) + \delta,$$

i.e. such that the definition of DP for the family $\{\nu_{\mathcal{D}} : \mathcal{D} \in \mathcal{S}\}$ is violated. Now, applying the inequality (17) followed by the inequality above we find

$$\begin{aligned} \mu_{\mathcal{D}'}(B) &\geqslant \nu_{\mathcal{D}'}(B) - \beta \\ &\geqslant e^{\varepsilon} \nu_{\mathcal{D}}(B) + \delta - \beta. \end{aligned}$$

Then, we apply inequality (16) to find

$$\mu_{\mathcal{D}'}(B) \geqslant e^{\varepsilon} \mu_{\mathcal{D}}(B) + \delta - (1 + e^{\varepsilon})\beta,$$

which proves the result. $\qquad\square$

## A.3. General result for Proposition 3.4

**Proposition A.4.** *Let $\{\mu_\mathcal{D} : \mathcal{D} \in \mathcal{S}\}$ and $\{\nu_\mathcal{D} : \mathcal{D} \in \mathcal{S}\}$ be two families of probability distributions. Assume $\{\nu_\mathcal{D} : \mathcal{D} \in \mathcal{S}\}$ is $(\varepsilon, \delta_\nu)$-differentially private and $\{\mu_\mathcal{D} : \mathcal{D} \in \mathcal{S}\}$ is not $(\varepsilon, \delta_\mu)$-DP for $\delta_\mu > \delta_\nu$, i.e. there exists at least one pair of adjacent datasets $\mathcal{D}, \mathcal{D}' \in \mathcal{S}$ and a measurable set B such that*

$$\mu_{\mathcal{D}'}(\mathsf{B}) > e^\varepsilon \mu_\mathcal{D}(\mathsf{B}) + \delta_\mu \ . \tag{18}$$

*Define the set of pairs of adjacent datasets for which (18) holds:*

$$\overline{\mathcal{S}} := \{\{\mathcal{D}, \mathcal{D}'\} \in \mathcal{S} \times \mathcal{S} : \mathcal{D}, \mathcal{D}' \text{ are adjacent and there exists B such that (18) holds}\}.$$

*Then, for any $\{\mathcal{D}, \mathcal{D}'\} \in \overline{\mathcal{S}}$ there exists $\tilde{\mathcal{D}} \in \{\mathcal{D}, \mathcal{D}'\}$ such that*

$$\|\nu_{\tilde{\mathcal{D}}} - \mu_{\tilde{\mathcal{D}}}\|_{\mathrm{TV}} > \frac{e^{-\varepsilon}}{1 + e^{-\varepsilon}}(\delta_\mu - \delta_\nu) \ .$$

*Proof.* We take any pair of datasets $\{\mathcal{D}, \mathcal{D}'\} \in \overline{\mathcal{S}}$. Notice that (18) implies that

$$\mu_\mathcal{D}(\mathsf{B}) < e^{-\varepsilon} \mu_{\mathcal{D}'}(\mathsf{B}) - e^{-\varepsilon}\delta_\mu \ .$$

Moreover, $(\varepsilon, \delta_\nu)$-DP of $\nu$ means that

$$\nu_\mathcal{D}(\mathsf{B}) \geqslant e^{-\varepsilon} \nu_{\mathcal{D}'}(\mathsf{B}) - e^{-\varepsilon}\delta_\nu \ .$$

Using both inequalities we find

$$\mu_\mathcal{D}(\mathsf{B}) - \nu_\mathcal{D}(\mathsf{B}) < -(e^{-\varepsilon}\delta_\mu - e^{-\varepsilon}\delta_\nu) + e^{-\varepsilon}\mu_{\mathcal{D}'}(\mathsf{B}) - e^{-\varepsilon}\nu_{\mathcal{D}'}(\mathsf{B})$$
$$< -e^{-\varepsilon}(\delta_\mu - \delta_\nu) + e^{-\varepsilon}(\mu_{\mathcal{D}'}(\mathsf{B}) - \nu_{\mathcal{D}'}(\mathsf{B})) \ .$$

Suppose now that $\|\mu_{\mathcal{D}'} - \nu_{\mathcal{D}'}\|_{\mathrm{TV}} \leqslant \zeta$ for some constant $0 \leqslant \zeta \leqslant \delta_\mu - \delta_\nu$. This implies that $\mu_{\mathcal{D}'}(\mathsf{B}) - \nu_{\mathcal{D}'}(\mathsf{B}) \leqslant \zeta$. Therefore, we find

$$\mu_\mathcal{D}(\mathsf{B}) - \nu_\mathcal{D}(\mathsf{B}) < -e^{-\varepsilon}(\delta_\mu - \delta_\nu - \zeta) \ .$$

Since $\delta_\mu - \delta_\nu - \zeta \geqslant 0$ by construction, taking the absolute value we find

$$|\mu_\mathcal{D}(\mathsf{B}) - \nu_\mathcal{D}(\mathsf{B})| > e^{-\varepsilon}(\delta_\mu - \delta_\nu - \zeta) \ .$$

By the definition of TV distance

$$\|\mu_\mathcal{D} - \nu_\mathcal{D}\|_{\mathrm{TV}} \geqslant |\mu_\mathcal{D}(\mathsf{B}) - \nu_\mathcal{D}(\mathsf{B})| > e^{-\varepsilon}(\delta_\mu - \delta_\nu - \zeta) \ .$$

Clearly, when $\|\mu_{\mathcal{D}'} - \nu_{\mathcal{D}'}\|_{\mathrm{TV}} \leqslant \zeta$ does not hold, it must be that $\|\mu_{\mathcal{D}'} - \nu_{\mathcal{D}'}\|_{\mathrm{TV}} > \zeta$. This shows that there exists a dataset $\tilde{\mathcal{D}} \in \{\mathcal{D}, \mathcal{D}'\}$ such that either $\|\mu_{\tilde{\mathcal{D}}} - \nu_{\tilde{\mathcal{D}}}\|_{\mathrm{TV}} > \zeta$ or $\|\mu_{\tilde{\mathcal{D}}} - \nu_{\tilde{\mathcal{D}}}\|_{\mathrm{TV}} > e^{-\varepsilon}(\delta_\mu - \delta_\nu - \zeta)$. Therefore it always holds that

$$\|\mu_{\tilde{\mathcal{D}}} - \nu_{\tilde{\mathcal{D}}}\|_{\mathrm{TV}} > \min\{\zeta, e^{-\varepsilon}(\delta_\mu - \delta_\nu - \zeta)\} \ .$$

Optimising the bound for $\zeta \in [0, \delta_\mu - \delta_\nu]$, we find that the lower bound above is maximised for $\zeta^* = \frac{e^{-\varepsilon}}{1+e^{-\varepsilon}}(\delta_\mu - \delta_\nu)$, which satisfies the equation $\zeta^* = e^{-\varepsilon}(\delta_\mu - \delta_\nu - \zeta^*)$. $\square$

## A.4. General result for Proposition 3.7

Consider the randomised algorithm

$$\mathcal{A}(\mathcal{D}) = g_\mathcal{D} + L, \tag{19}$$

where $g_\mathcal{D}$ is a random variable that depends on the dataset, and $L$ satisfies Assumption 3.5 and is independent of $g_\mathcal{D}$. We make the following assumptions, which is the general version of Assumption 3.6.

**Assumption A.5.** Let $\mathcal{D}, \mathcal{D}' \in \mathcal{S}$ denote two adjacent datasets. There exist $\eta > 0, \tilde{\delta} < 1$ that are independent of $\mathcal{D}, \mathcal{D}'$ such that

$$\mathbb{P}\left(|g_{\mathcal{D}} - g_{\mathcal{D}'}| \leqslant \eta\right) \geqslant 1 - \tilde{\delta} \,.$$

Assumption 3.5 and Assumption A.5 are enough to obtain the following guarantee on the differential privacy of the randomised algorithm (19).

**Proposition A.6.** *Let $\mathcal{A}$ be the randomised mechanism defined in (19), where $L$ is independent of $g_{\mathcal{D}}$ for any $\mathcal{D} \in \mathcal{S}$. Suppose Assumption A.5 holds, as well as Assumption 3.5 for $\eta$ as in Assumption A.5. Then $\mathcal{A}$ is $(\varepsilon, \tilde{\delta} + \delta)$-differentially private.*

*Proof.* Let B be any measurable set. We find

$$\mathbb{P}\left(\mathcal{A}(\mathcal{D}) \in \mathsf{B}\right) = \mathbb{P}\left(\mathcal{A}(\mathcal{D}) \in \mathsf{B}, |g_{\mathcal{D}} - g_{\mathcal{D}'}| \leqslant \eta\right) + \mathbb{P}\left(\mathcal{A}(\mathcal{D}) \in \mathsf{B}, |g_{\mathcal{D}} - g_{\mathcal{D}'}| > \eta\right). \tag{20}$$

The first term on the right hand side of (20) can be bounded using Assumption 3.5. Indeed, Assumption 3.5 can be used since we are on the event $|g_{\mathcal{D}} - g_{\mathcal{D}'}| \leqslant \eta$ and because $L$ and $g_{\mathcal{D}}, g_{\mathcal{D}'}$ are independent. Following this reasoning, we find

$$\mathbb{P}\left(g_{\mathcal{D}} + L \in \mathsf{B}, |g_{\mathcal{D}} - g_{\mathcal{D}'}| \leqslant \eta\right) \leqslant e^{\varepsilon} \mathbb{P}\left(g_{\mathcal{D}'} + L \in \mathsf{B}, |g_{\mathcal{D}} - g_{\mathcal{D}'}| \leqslant \eta\right) + \delta$$
$$\leqslant e^{\varepsilon} \mathbb{P}(\mathcal{A}(\mathcal{D}') \in \mathsf{B}) + \delta.$$

In the last inequality we simply discarded the event $\{|g_{\mathcal{D}} - g_{\mathcal{D}'}| \leqslant \eta\}$. Applying Assumption A.5 we can bound the second term in (20) as follows:

$$\mathbb{P}\left(\mathcal{A}(\mathcal{D}) \in \mathsf{B}, |g_{\mathcal{D}} - g_{\mathcal{D}'}| > \eta\right) \leqslant \mathbb{P}\left(|g_{\mathcal{D}} - g_{\mathcal{D}'}| > \eta\right) \leqslant \tilde{\delta}.$$

We have obtained the result. $\qquad \square$

### A.5. Proof of Proposition 3.10

Let $\mathcal{D}, \mathcal{D}' \in \mathcal{S}$ be two adjacent datasets. We introduce a joint process $(X_n^{\mathcal{D}}, X_n^{\mathcal{D}'})_{n=1}^N$ such that $(X_n^{\mathcal{D}})_{n=1}^N$ and $(X_n^{\mathcal{D}'})_{n=1}^N$ are two independent Markov chains respectively with transition kernels $P_{\mathcal{D}}$ and $P_{\mathcal{D}'}$, and initial distributions $\nu_{\mathcal{D}}, \nu_{\mathcal{D}'}$. We now define two sets, $\mathsf{B}_1$ and $\mathsf{B}_2$, that represent respectively Assumption 3.8 and Assumption 3.6:

$$\mathsf{B}_1 := \left\{ \left| \frac{1}{N} \sum_{n=1}^N f(X_n^{\tilde{\mathcal{D}}}) - \pi_{\tilde{\mathcal{D}}}(f) \right| \leqslant C_{\tilde{\delta}, N}, \text{ for } \tilde{\mathcal{D}} = \mathcal{D}, \mathcal{D}' \right\}$$

and

$$\mathsf{B}_2 := \left\{ \left| \frac{1}{N} \sum_{n=1}^N f(X_n^{\mathcal{D}}) - \frac{1}{N} \sum_{n=1}^N f(X_n^{\mathcal{D}'}) \right| \leqslant 2C_{\tilde{\delta}, N} + \gamma_f \right\}.$$

Now notice that $\mathsf{B}_1 \subset \mathsf{B}_2$ under our assumptions, and hence $\mathbb{P}(\mathsf{B}_2) \geqslant \mathbb{P}(\mathsf{B}_1)$. Indeed, an application of Assumption 3.9 gives

$$\left| \frac{1}{N} \sum_{n=1}^N f(X_n^{\mathcal{D}}) - \frac{1}{N} \sum_{n=1}^N f(X_n^{\mathcal{D}'}) \right| \leqslant \left| \frac{1}{N} \sum_{n=1}^N f(X_n^{\mathcal{D}}) - \pi_{\mathcal{D}}(f) \right| + |\pi_{\mathcal{D}}(f) - \pi_{\mathcal{D}'}(f)| + \left| \frac{1}{N} \sum_{n=1}^N f(X_n^{\mathcal{D}'}) - \pi_{\mathcal{D}'}(f) \right|$$
$$\leqslant 2C_{\tilde{\delta}, N} + \gamma_f \,.$$

Then, notice that, since the two chains are independent, we find

$$\mathbb{P}(\mathsf{B}_1) = \mathbb{P}\left( \left| \frac{1}{N} \sum_{n=1}^N f(X_n^{\mathcal{D}}) - \pi_{\mathcal{D}}(f) \right| \leqslant C_{\tilde{\delta}, N} \right) \mathbb{P}\left( \left| \frac{1}{N} \sum_{n=1}^N f(X_n^{\mathcal{D}'}) - \pi_{\mathcal{D}'}(f) \right| \leqslant C_{\tilde{\delta}, N} \right).$$

Hence, by Assumption 3.8

$$\mathbb{P}(\mathsf{B}_1) \geqslant (1 - \tilde{\delta})^2 = 1 - (2\tilde{\delta} - \tilde{\delta}^2).$$

Therefore, we have obtained $\mathbb{P}(\mathsf{B}_2) \geqslant 1 - (2\tilde{\delta} - \tilde{\delta}^2)$. The result then follows by Proposition A.6.

# B. Proofs and auxiliary results for Section 4

### B.1. Proof of Lemma 4.1

Consider a measurable set $B \subset \mathcal{X}$ and denote $X = \mathcal{A}(\mathcal{D})$, $Y = \mathcal{A}(\mathcal{D}')$. Let us introduce the set where the Radon-Nikodym derivative is larger than a constant $e^{\varepsilon}$ as

$$\mathsf{K}_\varepsilon := \left\{ \omega : \frac{\mathrm{d}\mathbb{P}}{\mathrm{d}\mathbb{Q}}(\omega) > e^\varepsilon \right\}.$$

Then we can decompose $\mathbb{P}(X \in B)$ as

$$\mathbb{P}(X \in B) = \mathbb{E}[\mathbb{1}_{X \in B} \mathbb{1}_{\mathsf{K}_\varepsilon}] + \mathbb{E}[\mathbb{1}_{X \in B} \mathbb{1}_{\mathsf{K}_\varepsilon^c}].$$

Hence, since by assumption we can control the Radon-Nikodym derivative with high probability, in the sense that $\mathbb{P}(\mathsf{K}_\varepsilon) \leqslant \delta$, we find that

$$\mathbb{E}[\mathbb{1}_{X \in B} \mathbb{1}_{\mathsf{K}_\varepsilon}] \leqslant \delta.$$

On the other hand, we have

$$\mathbb{E}[\mathbb{1}_{X \in B} \mathbb{1}_{\mathsf{K}_\varepsilon^c}] = \mathbb{E}_{\mathbb{Q}}\left[\mathbb{1}_{X \in B} \mathbb{1}_{\mathsf{K}_\varepsilon^c} \times \frac{\mathrm{d}\mathbb{P}}{\mathrm{d}\mathbb{Q}}\right] \leqslant e^\varepsilon \mathbb{E}_{\mathbb{Q}}\left[\mathbb{1}_{X \in B}\right] = e^\varepsilon \mathbb{P}(Y \in B),$$

as required.

### B.2. Proof of Lemma 4.2

By the data processing inequality (Theorem 9, (van Erven & Harremos, 2014)) one has

$$\mathsf{D}_\alpha(P_{\mathcal{A}(D)}^{\mathbb{P}} \| P_{\mathcal{A}(D')}^{\mathbb{P}}) = \mathsf{D}_\alpha(P_{\mathcal{A}(D)}^{\mathbb{P}} \| P_{\mathcal{A}(D)}^{\mathbb{Q}}) \leqslant \mathsf{D}_\alpha(\mathbb{P} \| \mathbb{Q}).$$

The result then follows by the definition of Rényi DP in (2) since

$$\mathbb{E}_{\mathbb{Q}}\left[\left(\frac{\mathrm{d}\mathbb{P}}{\mathrm{d}\mathbb{Q}}\right)^\alpha\right] = \mathbb{E}\left[\left(\frac{\mathrm{d}\mathbb{P}}{\mathrm{d}\mathbb{Q}}\right)^{\alpha-1}\right].$$

### B.3. Girsanov's theorem for SDEs

Here we apply Girsanov's theorem to obtain the Radon-Nikodym derivative of the law of two SDEs with different drifts. The precise formulation we choose here is chosen so as to be able to apply the result to the diffusion (7) featured in Section 4.2. We shall assume Novikov's condition (21).

**Lemma B.1.** *Consider two processes $X_t^i$ for $i = 1, 2$ on a filtered probability space $(\Omega, \mathcal{F}, (\mathcal{F}_t)_{t \geqslant 0}, \mathbb{P})$ that solve the SDEs*

$$\mathrm{d}X_t^1 = g(X_{\kappa(t)}^1, \eta_{\kappa(t)})\mathrm{d}t + \sqrt{2/\beta}\mathrm{d}W_t,$$

$$\mathrm{d}X_t^2 = g(X_{\kappa(t)}^2, \eta_{\kappa(t)})\mathrm{d}t + u_t\mathrm{d}t + \sqrt{2/\beta}\mathrm{d}W_t.$$

*where $X_0^1, X_0^2 = x_0 \in \mathbb{R}^d$, $(W_t)_{t \geqslant 0}$ is a standard Brownian motion, $g : \mathbb{R}^d \to \mathbb{R}^d$ is a function, $\kappa : [0, \infty) \to [0, \infty)$ satisfies $\kappa(t) \leqslant t$, $(\eta_t)_{t \geqslant 0}$ is a stochastic process independent of $(W_t)_{t \geqslant 0}$ and $u_t$ is a $\mathcal{F}_t$-adapted process. Let the SDE given for each $i = 1, 2$ have a unique strong solution, and suppose for $T > 0$ that*

$$\mathbb{E}\left[\exp\left(\frac{1}{2}\int_0^T |u_t|^2 \,\mathrm{d}t\right)\right] < \infty. \tag{21}$$

*Then there exists a measure $\mathbb{Q}$ such that the random variable $(X_t^1)_{t \in [0,T]}$ on the probability space $(\Omega, \mathcal{F}, (\mathcal{F}_t)_{t \geqslant 0}, \mathbb{Q})$ is equal in law to $(X_t^2)_{t \in [0,T]}$ on the probability space $(\Omega, \mathcal{F}, (\mathcal{F}_t)_{t \geqslant 0}, \mathbb{P})$, and furthermore*

$$\frac{\mathrm{d}\mathbb{Q}}{\mathrm{d}\mathbb{P}} = \exp\left(\sqrt{\frac{\beta}{2}}\int_0^T \langle u_s, \mathrm{d}W_s \rangle - \frac{\beta}{4}\int_0^T |u_s|^2 \mathrm{d}s\right). \tag{22}$$

*Proof.* For $t \geqslant 0$ let us define the new process

$$\widetilde{W}_t := \sqrt{\frac{\beta}{2}} \int_0^t u_s \mathrm{d}s + W_t,$$

We may now use Girsanov's theorem to find a measure $\mathbb{Q}$ such that $(\widetilde{W}_t)_{t \in [0,T]}$ is a Brownian motion under $\mathbb{Q}$. Particularly, under (21) one has by Corollary 5.13, Section 3.5 of Karatzas & Shreve (1991) that

$$\exp\left( \sqrt{\frac{\beta}{2}} \int_0^t \langle u_s, \mathrm{d}W_s \rangle - \frac{\beta}{4} \int_0^t |u_s|^2 \mathrm{d}s \right),$$

is a $(\mathcal{F}_t)_{t \geqslant 0}$-martingale under $\mathbb{P}$, so by Girsanov's theorem, i.e. Theorem 5.1, Section 3.5 of Karatzas & Shreve (1991), one has that $(\widetilde{W}_t)_{t \in [0,T]}$ is a Brownian motion under $\mathbb{Q}$ given by (22). Then since one may write

$$\mathrm{d}X_t^1 = g(X_{\kappa(t)}^1, \eta_{\kappa(t)})\mathrm{d}t + \sqrt{2/\beta}\mathrm{d}\widetilde{W}_t, \quad X_0^1 = x_0 \in \mathbb{R}^d,$$

and since both SDEs have unique strong solution one sees that $(X_t^1)_{t \in [0,T]}$ defined on the probability space $(\Omega, \mathcal{F}, (\mathcal{F}_t)_{t \geqslant 0}, \mathbb{Q})$ must be equal in law to $(X_t^2)_{t \in [0,T]}$ defined on $(\Omega, \mathcal{F}, (\mathcal{F}_t)_{t \geqslant 0}, \mathbb{P})$.

$\square$

### B.4. Bounds on Stochastic Integrals

In this section we present two well-known bounds for exponentials of stochastic integrals.

**Lemma B.2.** *Let $u_t$ be a $(\mathcal{F}_t)_{t \geqslant 0}$-measurable process such that for some $T, K > 0$ one has $\sup_{t \in [0,T]} |u_t| \leqslant K$. Then*

$$\mathbb{E}\left[ \exp\left( \int_0^T \langle u_s, \mathrm{d}W_s \rangle \right) \right] \leqslant e^{TK^2/2}.$$

*Proof.* Since we have that $|u_t| \leqslant K$ almost surely, one sees that for every $t \geqslant 0$

$$\mathbb{E}\left[ \exp\left( \frac{1}{2} \int_0^t |u_s|^2 \mathrm{d}s \right) \right] < \infty.$$

Therefore by Corollary 5.13, Section 3.5 of Karatzas & Shreve (1991) one has that

$$\exp\left( \int_0^t \langle u_s, \mathrm{d}W_s \rangle - \frac{1}{2} \int_0^t |u_s|^2 \mathrm{d}s \right),$$

is a martingale. Then using the assumed bound $|u_t| \leqslant K$ again one has

$$\mathbb{E}\left[ \exp\left( \int_0^T \langle u_s, \mathrm{d}W_s \rangle - TK^2/2 \right) \right] \leqslant \mathbb{E}\left[ \exp\left( \int_0^T \langle u_s, \mathrm{d}W_s \rangle - \frac{1}{2} \int_0^T |u_s|^2 \mathrm{d}s \right) \right] = 1,$$

at which point the result follows.

$\square$

**Lemma B.3.** *For $u_t$ as in Lemma B.2, one has for any $a > 0$*

$$\mathbb{P}\left( \int_0^T \langle u_s, \mathrm{d}W_s \rangle > a \right) \leqslant \exp\left( -\frac{a^2}{2TK^2} \right).$$

*Proof.* Let $v > 0$ and $a \in \mathbb{R}$. By Markov's inequality

$$\mathbb{P}\left( \int_0^T \langle u_s, \mathrm{d}W_s \rangle > a \right) = \mathbb{P}\left( \exp\left( v \int_0^T \langle u_s, \mathrm{d}W_s \rangle \right) > e^{av} \right) \leqslant e^{-av}\mathbb{E}\left[ \exp\left( v \int_0^T \langle u_s, \mathrm{d}W_s \rangle \right) \right],$$

so that applying Lemma B.2 one has

$$\mathbb{P}\left( \int_0^T \langle u_s, \mathrm{d}W_s \rangle > a \right) \leqslant e^{-av + Tv^2K^2/2}.$$

The result follows by minimising $v$.

$\square$

## B.5. Proof of Proposition 4.6

Let us fix adjacent datasets $\mathcal{D}, \mathcal{D}' \in \mathcal{S}$. Then by the assumptions and the version of Girsanov's theorem presented in Lemma B.1, one sees that there exists a measure $\mathbb{Q}$ under which $(X_t^{\mathcal{D}})_{t \in [0,T]}$ is equal in law to $(X_t^{\mathcal{D}'})_{t \in [0,T]}$ under $\mathbb{P}$, that is

$$P_{(X_t^{\mathcal{D}})_{t \in [0,T]}}^{\mathbb{Q}} = P_{(X_t^{\mathcal{D}'})_{t \in [0,T]}}^{\mathbb{P}},$$

and furthermore

$$\frac{\mathrm{d}\mathbb{Q}}{\mathrm{d}\mathbb{P}} = \exp\left( \sqrt{\frac{\beta}{2}} \int_0^T \langle h(X_{\kappa(t)}^{\mathcal{D}}, \eta_{\kappa(t)}), \mathrm{d}W_t \rangle - \frac{\beta}{4} \int_0^T |h(X_{\kappa(t)}^{\mathcal{D}}, \eta_{\kappa(t)})|^2 \mathrm{d}t \right),$$

for

$$h(x, \eta) := f(x, \eta, \mathcal{D}) - f(x, \eta, \mathcal{D}').$$

Then, Lemmas 4.1 and 4.2 give that DP and Rényi-DP can be obtained by controlling respectively the tails and the moments of $\frac{\mathrm{d}\mathbb{P}}{\mathrm{d}\mathbb{Q}}$.

Let us first focus on the DP bound. For any $\delta > 0$, let $\varepsilon_\delta := Tc^2\beta/4 + c\sqrt{\beta T \log(1/\delta)}$. Since $|h| \leqslant c$ by assumption, one has

$$\mathbb{P}\left( \frac{\mathrm{d}\mathbb{P}}{\mathrm{d}\mathbb{Q}} > e^{\varepsilon_\delta} \right) \leqslant \mathbb{P}\left( \exp\left( -\sqrt{\frac{\beta}{2}} \int_0^T \langle h(X_{\kappa(t)}^{\mathcal{D}}, \eta_{\kappa(t)}), \mathrm{d}W_t \rangle \right) > \exp\left( \varepsilon_\delta - \frac{\beta}{4} \int_0^T |h(X_{\kappa(t)}^{\mathcal{D}}, \eta_{\kappa(t)})|^2 \mathrm{d}t \right) \right)$$

$$\leqslant \mathbb{P}\left( \exp\left( -\sqrt{\frac{\beta}{2}} \int_0^T \langle h(X_{\kappa(t)}^{\mathcal{D}}, \eta_{\kappa(t)}), \mathrm{d}W_t \rangle \right) > \exp\left( \varepsilon_\delta - \frac{\beta}{4} Tc^2 \right) \right)$$

$$= \mathbb{P}\left( \int_0^T \langle -\sqrt{\frac{\beta}{2}} h(X_{\kappa(t)}^{\mathcal{D}}, \eta_{\kappa(t)}), \mathrm{d}W_t \rangle > c\sqrt{\beta T \log(1/\delta)} \right)$$

Now, applying Lemma B.3 for $u_t = -\sqrt{\frac{\beta}{2}} h(X_{\kappa(t)}^{\mathcal{D}}, \eta_{\kappa(t)})$, which under our assumptions satisfies $|u_t| \leqslant c\sqrt{\frac{\beta}{2}}$, we find

$$\mathbb{P}\left( \frac{\mathrm{d}\mathbb{P}}{\mathrm{d}\mathbb{Q}} > e^{\varepsilon_\delta} \right) \leqslant \delta.$$

Finally, Lemma 4.1 gives the wanted $(\varepsilon, \delta)$-DP bound.

Now we derive the bound on the Rényi-DP for $\alpha > 1$. Lemma 4.2 shows that it is sufficient to bound

$$\mathbb{E}\left[ \left( \frac{\mathrm{d}\mathbb{P}}{\mathrm{d}\mathbb{Q}} \right)^{\alpha-1} \right] = \mathbb{E}\left[ \exp\left( \frac{\beta(\alpha-1)}{4} \int_0^T |h(X_{\kappa(t)}^{\mathcal{D}}, \eta_{\kappa(t)})|^2 \mathrm{d}t - (\alpha-1)\sqrt{\frac{\beta}{2}} \int_0^T \langle h(X_{\kappa(t)}^{\mathcal{D}}, \eta_{\kappa(t)}), \mathrm{d}W_t \rangle \right) \right].$$

We can bound the first term using that $|h(X_{\kappa(t)}^{\mathcal{D}}, \eta_{\kappa(t)})| \leqslant c$, while the second term can be dealt applying Lemma B.2 for $u_t = (1-\alpha)(\beta/2)^{1/2} h(X_{\kappa(t)}^{\mathcal{D}}, \eta_{\kappa(t)})$. This gives

$$\mathbb{E}\left[ \left( \frac{\mathrm{d}\mathbb{P}}{\mathrm{d}\mathbb{Q}} \right)^{\alpha-1} \right] \leqslant \exp\left( \frac{\beta(\alpha-1)}{4} Tc^2 \right) \mathbb{E}\left[ \exp\left( \int_0^T \langle (1-\alpha)\sqrt{\frac{\beta}{2}} h(X_{\kappa(t)}^{\mathcal{D}}, \eta_{\kappa(t)}), \mathrm{d}W_t \rangle \right) \right]$$

$$\leqslant \exp\left( \frac{\beta}{4} Tc^2 \alpha(\alpha-1) \right).$$

By Lemma 4.2 this implies Rényi-DP with $\varepsilon = \frac{\beta}{4} Tc^2 \alpha$.

## B.6. Proof of Proposition 4.5

Let us assume $T \geqslant 1$, noting that the result holds by Proposition 4.6 for $T \leqslant 1$. Let us fix adjacent datasets $\mathcal{D}, \mathcal{D}' \in \mathcal{S}$ and let us define the process $(Z_t)_{t \in [0,T]}$ by

$$\mathrm{d}Z_t = f_{\mathcal{D}}(Z_{\kappa(t)}, \eta_{\kappa(t)}) \mathrm{d}t + u_t \mathrm{d}t + \sqrt{2/\beta} \mathrm{d}W_t, \quad Z_0 = x_0 \in \mathbb{R}^d,$$

where $u_t$ is supported on $t \in [T-1, T]$ and in particular

$$u_t := [f_{\mathcal{D}'}(X^{\mathcal{D}'}_{\kappa(t)}, \eta_{\kappa(t)}) - f_{\mathcal{D}}(Z_{\kappa(t)}, \eta_{\kappa(t)}) - (Z_{T-1} - X^{\mathcal{D}'}_{T-1})] \mathbb{1}_{t \in [T-1, T]}.$$

Then, for $t \in [T-1, T]$ one has

$$\begin{aligned} Z_t - X^{\mathcal{D}'}_t &= Z_{T-1} - X^{\mathcal{D}'}_{T-1} + \int_{T-1}^{t} \left( f_{\mathcal{D}}(Z_{\kappa(s)}, \eta_{\kappa(s)}) + u_s - f_{\mathcal{D}'}(X^{\mathcal{D}'}_{\kappa(s)}, \eta_{\kappa(s)}) \right) \mathrm{d}s \\ &= Z_{T-1} - X^{\mathcal{D}'}_{T-1} - \int_{T-1}^{t} (Z_{T-1} - X^{\mathcal{D}'}_{T-1}) \mathrm{d}s \\ &= (T-t) \left( Z_{T-1} - X^{\mathcal{D}'}_{T-1} \right), \end{aligned}$$

and hence $Z_T = X^{\mathcal{D}'}_T$ almost surely. Furthermore, since (7) has a unique strong solution for each dataset one has that $Z_t = X^{\mathcal{D}}_t$ for $t \in [0, T-1]$ almost surely. Therefore by assumption one has for $t \in [0, T-1]$ that

$$|Z_t - X^{\mathcal{D}'}_t| = |X^{\mathcal{D}}_t - X^{\mathcal{D}'}_t| \leqslant C.$$

Furthermore note that, for $t \in [T-1, T]$,

$$\begin{aligned} |Z_t - X^{\mathcal{D}'}_t| &= (T-t)|X^{\mathcal{D}}_{T-1} - X^{\mathcal{D}'}_{T-1}| \\ &\leqslant |X^{\mathcal{D}}_{T-1} - X^{\mathcal{D}'}_{T-1}| \\ &\leqslant C. \end{aligned}$$

Hence, for $t \in [0, T]$ one has

$$|Z_{\kappa(t)} - X^{\mathcal{D}'}_{\kappa(t)}| \leqslant C.$$

Using (8) one may therefore bound

$$|u_t| \leqslant L|X^{\mathcal{D}'}_{\kappa(t)} - Z_{\kappa(t)}| + c + C \leqslant C(L+1) + c.$$

Now observe that by Lemma B.1, under the measure $\mathbb{Q}$ given as

$$\mathrm{d}\mathbb{Q} = \exp\left( \sqrt{\frac{\beta}{2}} \int_{T-1}^{T} \langle u_t, \mathrm{d}W_t \rangle - \frac{\beta}{4} \int_{T-1}^{T} |u_t|^2 \mathrm{d}t \right) \mathrm{d}\mathbb{P},$$

one has that

$$P^{\mathbb{Q}}_{(Z_t)_{t \in [0,T]}} = P^{\mathbb{P}}_{(X^{\mathcal{D}}_t)_{t \in [0,T]}}. \tag{23}$$

Moreover, since $Z_T = X^{\mathcal{D}'}_T$ almost surely, by (23) one has that

$$P^{\mathbb{Q}}_{X^{\mathcal{D}'}_T} = P^{\mathbb{Q}}_{Z_T} = P^{\mathbb{P}}_{X^{\mathcal{D}}_T}.$$

Now, we wish to apply Lemmas 4.1 and 4.2 to obtain the wanted guarantees respectively on the DP and Rényi-DP of our randomised algorithm. For any $\delta > 0$ and $\alpha > 1$, let us set $\varepsilon_\delta = C_2/4 + \sqrt{C_2 \log(1/\delta)}$ and $\varepsilon_\alpha = \alpha C_2/4$, where $C_2 = \beta(C(L+1) + c)^2$. Proceeding as in the proof of Proposition 4.6 one may show that

$$\mathbb{P}\left( \frac{\mathrm{d}\mathbb{P}}{\mathrm{d}\mathbb{Q}} \geqslant \varepsilon_\delta \right) \leqslant \delta, \qquad \mathsf{D}_\alpha(\mathbb{P}\|\mathbb{Q}) \leqslant \varepsilon_\alpha,$$

which is sufficient to apply Lemmas 4.1 and 4.2.

# C. Proofs for Section 5

## C.1. Proof of Theorem 5.2

Let us fix adjacent datasets $\mathcal{D}, \mathcal{D}' \in \mathcal{S}$ and consider two versions $(x_n^{\mathcal{D}})_{n \geqslant 1}$ and $(x_n^{\mathcal{D}'})_{n \geqslant 1}$ of the ULA algorithm (12) targeting $\pi_{\mathcal{D}}$ and $\pi_{\mathcal{D}'}$ respectively. Let $(x_n^{\mathcal{D}})_{n \geqslant 1}$ and $(x_n^{\mathcal{D}'})_{n \geqslant 1}$ be driven by the same sequence $(z_n)_{n \geqslant 1}$ of iid standard Gaussians. Let us define

$$e_n := x_n^{\mathcal{D}} - x_n^{\mathcal{D}'},$$

so that by the strong monotonicity and $L$-Lipschitz assumption on $\nabla K$ one has

$$|e_n - \gamma \nabla K(x_n^{\mathcal{D}}) - \gamma \nabla K(x_n^{\mathcal{D}'})|^2 \leqslant (1 - 2\gamma a + \gamma^2 L^2)|e_n|^2.$$

Now let us show

$$0 \leqslant 1 - 2\gamma a + \gamma^2 L^2 \leqslant 1. \tag{24}$$

For the lower bound one can show by standard calculus that the minimum value attained for $\gamma > 0$ in the above expression is $1 - a^2/L^2$. To see that this is greater than 0, note that by the monotonicity assumption (13), applying Cauchy-Schwarz one must have

$$|\nabla K(x) - \nabla K(y)| \geqslant a|x - y|,$$

and therefore one must have $L \geqslant a$. The upper bound in (24) follows simply by the assumption $\gamma \in (0, 2a/L^2)$. Then since $\sqrt{1-x} \leqslant 1 - x/2$ for $x \in [0,1]$, one may conclude

$$|e_n - \gamma \nabla K(x_n^{\mathcal{D}}) - \gamma \nabla K(x_n^{\mathcal{D}'})| \leqslant (1 - \gamma a + \gamma^2 L^2/2)|e_n|.$$

By the assumption that that $|\nabla V|, |\nabla V'| \leqslant c$, one therefore has that

$$\begin{aligned}
|e_{n+1}| &\leqslant |e_n - \gamma \nabla K(x_n^{\mathcal{D}}) - \gamma \nabla K(x_n^{\mathcal{D}'})| + |\gamma \nabla V^{\mathcal{D}}(x_n^{\mathcal{D}}) - \gamma \nabla V^{\mathcal{D}'}(x_n^{\mathcal{D}'})| \\
&\leqslant (1 - \gamma a + \gamma^2 L^2/2)|e_n| + 2\gamma c.
\end{aligned}$$

Iterating this bound from $n = 0$ (since $e_0 = 0$) one obtains for all $n \geqslant 1$ that

$$|e_n| \leqslant \frac{2c}{a - \gamma L^2/2}. \tag{25}$$

Now let us consider the following interpolation of the ULA algorithm. Let us define $\kappa_\gamma : [0, \infty) \to [0, \infty)$ by $\kappa_\gamma(t) := \gamma \lfloor t/\gamma \rfloor$, that is, the projection backwards onto $\{0, \gamma, 2\gamma, ...\}$. Then consider the continuous time process

$$\mathrm{d}X_t^{\mathcal{D}} = -\nabla U_{\mathcal{D}}(X_{\kappa_\gamma(t)}^{\mathcal{D}})\mathrm{d}t + \sqrt{2}\mathrm{d}W_t, \quad X_0^{\mathcal{D}} = x_0 \in \mathbb{R}^d,$$

and likewise for $\mathcal{D}'$. It is easy to verify that since the coefficients of the drift are fixed in-between the time-discretisation grid $\{0, \gamma, 2\gamma, ...\}$, one has that the law of $X_{n\gamma}^{\mathcal{D}}$ is equal to the law of $(x_n^{\mathcal{D}})$, and likewise for $\mathcal{D}'$. Furthermore, since $X_T^{\mathcal{D}}$ and $X_t^{\mathcal{D}'}$ are driven by the same Brownian motion, one may apply (25) to conclude that

$$\sup_{t \geqslant 0} |X_{\kappa_\gamma(t)}^{\mathcal{D}} - X_{\kappa_\gamma(t)}^{\mathcal{D}'}| \leqslant \frac{2c}{a - \gamma L^2/2}.$$

So as to satisfy the assumptions of Proposition 4.5 we need to extend this bound from grid points $\kappa_\gamma(t)$ to the whole of $t \geqslant 0$. To see this note that

$$X_t^{\mathcal{D}} - X_t^{\mathcal{D}'} = X_{\kappa_\gamma(t)}^{\mathcal{D}} - X_{\kappa_\gamma(t)}^{\mathcal{D}'} - (t - \kappa_\gamma(t))[\nabla U_{\mathcal{D}}(X_{\kappa_\gamma(t)}^{\mathcal{D}}) - \nabla U_{\mathcal{D}'}(X_{\kappa_\gamma(t)}^{\mathcal{D}'})], \tag{26}$$

and furthermore for arbitrary $x, y \in \mathbb{R}^d$ and $t > 0$, if one defines

$$f(t) := |x + yt|^2 = |x|^2 + 2t\langle a, y \rangle + t^2|y|^2,$$

then since $f'' > 0$ the function $f$ cannot have a local maximum anywhere. Therefore

$$\sup_{t \in [a,b]} f(t) \leqslant \max\{f(a), f(b)\}.$$

Applying this principle to (26) one sees that

$$\sup_{t \geqslant 0} |X_t^{\mathcal{D}} - X_t^{\mathcal{D}'}| \leqslant \frac{2c}{a - \gamma L^2/2}.$$

Therefore one sees that the assumptions of Proposition 4.5 are satisfied for $L, c > 0$ as in the Proposition, and $C = \frac{2c}{a - \gamma L^2/2}$, and the result therefore follows by choosing $T = n\gamma$

## C.2. Proof of Theorem 5.6

As in the proof of Theorem 5.2, for adjacent data $\mathcal{D} = (d_1, ..., d_m), \mathcal{D}' = (d_1', ..., d_m')$ we show that $x_n^{\mathcal{D}}$ and $x_n^{\mathcal{D}'}$ are almost surely close. The strategy here is essentially the same. Letting $e_n := x_n^{\mathcal{D}} - x_n^{\mathcal{D}'}$ as before, one has

$$\left| e_n - \frac{\gamma}{s} \sum_{i \in A_{n+1}} (\nabla k(x_n^{\mathcal{D}}) - \nabla k(x_n^{\mathcal{D}'})) \right|^2 = |e_n|^2 - 2\frac{\gamma}{s} \sum_{i \in A_{n+1}} \langle e_n, \nabla k(x_n^{\mathcal{D}}) - \nabla k(x_n^{\mathcal{D}'}) \rangle$$
$$+ \left| \frac{\gamma}{s} \sum_{i \in A_{n+1}} (\nabla k(x_n^{\mathcal{D}}) - \nabla k(x_n^{\mathcal{D}'})) \right|^2,$$

so that since by the Lipschitz assumption and the fact $A_{n+1} \subset \{1, 2, ..., m\}$ is of size $s$, one obtains

$$\left| \frac{\gamma}{s} \sum_{i \in A_{n+1}} (\nabla k(x_n^{\mathcal{D}}) - \nabla k(x_n^{\mathcal{D}'})) \right| \leqslant \gamma L |e_n|.$$

Then applying the convexity assumption one obtains

$$|e_n - \frac{\gamma}{s} \sum_{i \in A_{n+1}} (\nabla k(x_n^{\mathcal{D}}) - \nabla k(x_n^{\mathcal{D}'}))|^2 \leqslant (1 - 2a + L^2\gamma^2)|e_n|^2, \tag{27}$$

so that since as before one has $0 \leqslant 1 - 2a + L^2\gamma^2 \leqslant 1$, one may square root to see that

$$|e_{n+1}| = |e_n - \frac{\gamma}{s} \sum_{i \in A_{n+1}} (\nabla k(x_n^{\mathcal{D}}) - \nabla k(x_n^{\mathcal{D}'}))|^2 + \frac{\gamma}{s} \sum_{i \in A_{n+1}} |\nabla_x v(x_n^{\mathcal{D}}, d_i) - \nabla_x v(x_n^{\mathcal{D}'}, d_i')|, \tag{28}$$

so that using the assumption $|\nabla_x v(\cdot, \cdot)| \leqslant c$ one sees that the bound (25) from before also holds in this setting.

Now let us find a continuous time interpolation of the SGD process (15). For the stochastic gradient we consider the process $(\eta_t)_{t \geqslant 0}$ given for $n \in \mathbb{N}$ and $t \in (n\gamma, (n+1)\gamma)$ as $\eta_t = A_{n+1}$, where $(A_n)_{n \geqslant 1}$ is the sequence of i.i.d. random variables uniformly distributed on subsets of size $s \leqslant m$ of $\{1, 2, ..., m\}$. Then for every data $\mathcal{D} = (x_1, ..., x_m)$ one may define the process $(Y_t^{\mathcal{D}})_{t \geqslant 0}$ as the solution to the SDE

$$dY_t^{\mathcal{D}} = -\frac{1}{s} \sum_{i \in \eta_t} \nabla_x \ell(Y_{\kappa_\gamma(t)}^{\mathcal{D}}, d_i) dt + \sqrt{2/\beta} dW_t, \tag{29}$$

where $\kappa_\gamma$ is the backwards projection to $\{0, \gamma, 2\gamma, ...\}$ as in the proof of Theorem 5.2. Then it is easy to verify that the law of $Y_{n\gamma}^{\mathcal{D}}$ is equal to the law of $x_n^{\mathcal{D}}$, at which point one can use the strategy from the proof of Theorem 5.2 to obtain that

$$\sup_{t \geqslant 0} |Y_t^{\mathcal{D}} - Y_t^{\mathcal{D}'}| \leqslant \frac{2c}{a - \gamma L^2/2},$$

at which point as before the result follows from Proposition 4.5, paying attention to the value of $\beta > 0$ assumed in the definition (15) of the SGD algorithm.

## C.3. Proof of Theorem 5.7

Here we modify slightly proof of Theorem 5.6 previously. Specifically, note that if $\mathcal{D} = (d_1, ..., d_m)$ and $\mathcal{D}' = (d_1', ..., d_m')$ are adjacent then only one of their entries is different, that is there exists $i \in \{1, 2, ..., m\}$ such that $d_i \neq d_i'$, however for

$j \in \{1, 2, ..., m\}$ such that $j \neq i$ one has $d_j = d'_j$. Therefore, since we have assumed $v_\theta(\cdot, d)$ to be constant for every $d \in \mathcal{D}$, one may bound

$$\frac{\gamma}{s} \sum_{i \in A_{n+1}} |\nabla_x v(x_n^{\mathcal{D}}, d_i) - \nabla_x v(x_n^{\mathcal{D}'}, d'_i)| \leqslant \frac{2c\gamma}{s},$$

and therefore using (27) and (28) one has that

$$|e_{n+1}| \leqslant (1 - \gamma a + \gamma^2 L^2/2)|e_n| + \frac{2c\gamma}{s},$$

and as a result

$$\sup_{n \geqslant 1} |e_n| \leqslant \frac{2c}{s(\gamma L^2/2 - \mathcal{Y})}.$$

Then extending this bound to the continuous interpolation (29) as in the proofs of Theorems 5.2 and 5.6, and applying Proposition 4.5 as before, the result follows.

## C.4. Proof of Theorem 5.8

The proof here is very simple: note that the assumptions imply that for every pair of adjacent datasets $\mathcal{D}, \mathcal{D}'$ and $x \in \mathbb{R}^d$, since $A_{n+1}$ is a subset of size $s \leqslant m$ of $\{1, 2, ..., m\}$, one has $\mathcal{D} = (d_1, ..., d_m)$, $\mathcal{D}' = (d'_1, ..., d'_m)$ that differ in only one element, so

$$\left| \frac{1}{s} \sum_{i \in \eta_t} \nabla_x \ell(x, d_i) - \frac{1}{s} \sum_{i \in \eta_t} \nabla_x \ell(x, d'_i) \right| \leqslant c/s.$$

Therefore one may apply Proposition 4.6 to (29), and then since the law of $(Y_0^{\mathcal{D}}, Y_\gamma^{\mathcal{D}}, ... Y_{n\gamma}^{\mathcal{D}})$ is equal to the law of $(x_0^{\mathcal{D}}, ..., x_n^{\mathcal{D}})$ and $(Y_0^{\mathcal{D}}, Y_\gamma^{\mathcal{D}}, ... Y_{n\gamma}^{\mathcal{D}})$ is equal to a mapping from $(Y_t)_{t \in [0, n\gamma]}$, the result follows from the data processing inequality (see Theorem 9 in van Erven & Harremos (2014)).

