# OpenReview forum: "Differential Privacy Guarantees of Markov Chain Monte Carlo Algorithms"
_ICML.cc/2025/Conference — ICML 2025 poster_

### Official Review · Reviewer_WFdZ · 2025-03-12

**Overall Recommendation:** 4

**Summary:**

This papers studies differential privacy
and R\'{e}nyi differential privacy
for Markov chain Monte Carlo (MCMC) algorithms
for the path and the final value of the algorithms.
The general results are then applied
to study two popular MCMC algorithms,
the unadjusted Langevin algorithm (ULA)
and stochastic gradient Langevin dynamics (SGLD).
The paper establishes the privacy
guarantees uniform in number of iterations, and the bounds
on the privacy on the privacy of the entire trajectory.
As a result, this answers an open question (Question 1.1. in Altschuler and Talwar (2022))
about uniform-in-time differential privacy guarantees
in a non-convex setting on an unbounded space.
The results generalize the results of Chourasia et al. (2021)
to a non-convex setting, and match theirs in the strongly convex regime.

**Claims And Evidence:**

Yes.

**Essential References Not Discussed:**

N.A.

**Experimental Designs Or Analyses:**

N.A.

**Methods And Evaluation Criteria:**

Yes.

**Other Comments Or Suggestions:**

1) In the very beginning of Section 3.2., do you need any assumption on $f$?

(2) In the last line in Lemma 4.2, it is better to write $(\alpha,\epsilon)$-R\'{e}nyi-DP instead.

(3) Currently, Proposition 4.5. is about privacy of the path, and Proposition 4.6.
is about privacy of the final iterate. However, when you present the corresponding results
for ULA for example, you present the results for the final iterate first (Theorem 5.2) and
before you present the results for the path (Theorem 5.4). I think it is better to make
the ordering consistent.

(4) In the first equation in Proposition 4.6., is it (uniformly) for every $s$? Please make it more clear.

**Other Strengths And Weaknesses:**

Strengths

(1) The paper is well written.

(2) The analysis seems to be rigorous.

(3) As the paper claims, the paper answers an open question (Question 1.1. in Altschuler and Talwar (2022))
about uniform-in-time differential privacy guarantees
in a non-convex setting on an unbounded space.

(4) The results generalize the results of Chourasia et al. (2021)
to a non-convex setting, and match theirs in the strongly convex regime.

Weaknesses

(1) The assumption (equation (8)) in Proposition 4.5. seems super strong to me.
As a result, when applied to ULA and SGLD, the Assumption 5.3.
is super strong.
Can you add some discussions on some real examples that satisfy such an assumption?

(2) Even though the results generalize the results of Chourasia et al. (2021)
to a non-convex setting, the non-convexity here seems to be quite restrictive, in the sense
that it is simply a strongly convex and smooth function plus a function whose gradient
is uniformly bounded.

**Questions For Authors:**

See my comments for weaknesses above.

**Relation To Broader Scientific Literature:**

The paper establishes the privacy
guarantees uniform in number of iterations, and the bounds
on the privacy on the privacy of the entire trajectory.
As a result, this answers an open question (Question 1.1. in Altschuler and Talwar (2022))
about uniform-in-time differential privacy guarantees
in a non-convex setting on an unbounded space.
The results generalize the results of Chourasia et al. (2021)
to a non-convex setting, and match theirs in the strongly convex regime.

**Theoretical Claims:**

The paper seems to be rigorous though I don't have to check all the proofs details.

---

> ### Author Rebuttal · Authors · 2025-03-31
>
> We are grateful to the reviewer for the positive review and for their comments. We will incorporate the reviewer's suggestions in the revised version of the paper. Below we reply to the questions that the reviewer raised:
>
> - **In the very beginning of Section 3.2., do you need any assumption on $f$?**
>   We do not directly need to state assumptions on $f$ straight away, but restrictions on $f$ indirectly come into play to ensure that the assumptions that follow are satisfied. We will add a comment about this in the revised version of the paper.
>
> - **In the last line in Lemma 4.2, it is better to write $(\alpha,\varepsilon)$-Rényi-DP instead.**
>   We agree and will correct this.
>
> - **Currently, Proposition 4.5. is about privacy of the path, and Proposition 4.6. is about privacy of the final iterate...**
>   We agree with the reviewer and we will incorporate this change in the revised paper.
>
> - **In the first equation in Proposition 4.6., is it (uniformly) for every $s$? Please make it more clear.**
>   Indeed, here we forgot to mention that the result should hold uniformly over $s$, as well as for any $x,y\in \mathbb{R}^d$. This will be fixed in the revised paper.
>
> - Regarding the weaknesses of the paper: we agree that the assumptions we consider are quite restrictive, but these should be compared to the existing literature. It is likely that relatively strict structural assumptions are needed in any case for the class of algorithms we consider. Regardless, we are going to add a discussion about the assumptions as suggested, highlighting their weaknesses and when they might be satisfied.

---

### Official Review · Reviewer_kjX6 · 2025-03-14

**Overall Recommendation:** 3

**Summary:**

This paper analyzes the differential privacy (DP) guarantees of Markov Chain Monte Carlo (MCMC) algorithms, focusing on both general MCMC methods and specific Langevin-based variants. It establishes that the DP properties of the posterior distribution are crucial for ensuring the privacy of MCMC samples, showing that if the posterior is not differentially private, the MCMC chain cannot be either. Using Girsanov’s theorem and a perturbation technique, the authors derive DP and Rényi DP bounds for the entire trajectory and final iterates of the Unadjusted Langevin Algorithm (ULA) and Stochastic Gradient Langevin Dynamics (SGLD). Their results improve on standard composition bounds by providing uniform-in-time privacy guarantees, particularly in non-convex settings, which were previously challenging.

**Claims And Evidence:**

See questions

**Essential References Not Discussed:**

No

**Experimental Designs Or Analyses:**

NA

**Methods And Evaluation Criteria:**

NA

**Other Comments Or Suggestions:**

- Proposition 4.6. Is "for any x, y, s..." missing?

**Other Strengths And Weaknesses:**

See questions

**Questions For Authors:**

- The authors claim that "In particular, we find that the DP of the posterior is the crucial starting point to perform Bayesian inference. This fact is supported by Proposition 3.4, which shows that if the posterior has weaker DP guarantees than the MCMC algorithm, the law of the MCMC chain after n iterations is far from the posterior in total variation distance." However, in Proposition 3.4, the $\pi$ is not defined as the stationary distribution of the transition kernel, which may leave a gap between Proposition 3.4 and the claim on the DP and convergence rate. Could the authors provide an example of MCMC that satisfies Proposition 3.4 (since it is a lower bound)?

- Can the authors discuss the connection between Assumption 5.1 and the log-Sobolev inequality? Section 5 considers the non-convex setting. Does the function $K$ serve as a strongly convex regularization? Could the author explain how it is used in deriving the results for algorithms where discretization is needed?

- In Proposition 4.6 (Line 283) and Line 865, could the author specify: "almost surely" with respect to what probability measure?

- Could the author provide intuition on how this path-wise analysis is better than the advanced composition theorem of DP? Does the discretization reintroduce DP accounting similar to the composition-style analysis? Can the author further clarify how the resulting order is better than previous results?

**Relation To Broader Scientific Literature:**

NA

**Theoretical Claims:**

Yes. I checked the proof of Lemma 4.1. See questions

---

> ### Author Rebuttal · Authors · 2025-03-31
>
> We thank the reviewer for their useful comments and remarks. We are going to incorporate them in the revised version of the paper. Below we address the questions from the "Questions for authors" section.
>
> - **Proposition 4.6. Is "for any x, y, s..." missing?**
>   Indeed, we will add this in the revised paper.
>
> - **The authors claim that "In particular, we find that the DP of the posterior is the crucial starting point to perform ..."**
>   We shall clarify this aspect in the revised version of the paper.
>   Indeed, in Proposition 3.4 we do not make an assumption on the stationary distribution of the Markov chain. This is because such an assumption is not required to obtain the result, and moreover, our statement then applies also to biased MCMC algorithms such as the unadjusted Langevin algorithm. In essence, the result states that the law of a differentially private MCMC algorithm can be far from any probability distribution that has worse DP guarantees. In our context, we are particularly interested in the distance to the posterior distribution, hence we state the result in such a form. The Proposition then applies very broadly to any MCMC algorithm with a DP guarantee after $n$-steps that is not satisfied by the posterior distribution.
>
> - **Can the authors discuss the connection between Assumption 5.1 and the log-Sobolev inequality? Section 5 considers the non-convex setting. Does the function serve as a strongly convex regularization? Could the authors explain how it is used in deriving the results for algorithms where discretization is needed?**
>   The reviewer brings up a very interesting connection with structural properties of the target, one which we are interested in exploring in future works. It is true that under Assumption 5.1 the target density does obey a log-Sobolev inequality. However, it would not be possible to prove privacy bounds solely under the assumption that for each $\mathcal{D}$ the posterior $\pi_\mathcal{D}$ obeys a log-Sobolev inequality, since this is not enough to ensure the densities are close (for instance, by shifting the target densities).
>   The assumption of a strongly convex part $K$ of the target posterior could indeed be a strongly convex regularizer, either for a Bayesian prior or a loss function, and we shall mention this in the revised version. We write $U_{\mathcal{D}}=K+V_{\mathcal{D}}$ since the former part induces a contraction and the latter part induces the processes to move apart. This is the same for the discretization and for continuous-time processes which target $\pi_\mathcal{D}$.
>
> - **In Proposition 4.6 (Line 283) and Line 865, could the authors specify: "almost surely" with respect to what probability measure?**
>   Thank you for bringing this to our attention, we will clarify this aspect. The distinction between being almost surely true for $\mathbb{P}$ or $\mathbb{Q}$ is not significant since the measures are absolutely continuous and therefore have the same null sets (this is always true for measures constructed via Girsanov's theorem). We shall make this clearer in the revised version. More generally, we only consider randomness induced by the algorithm, while the dataset $\mathcal{D}$ is always considered fixed.
>
> - **Could the authors provide intuition on how this path-wise analysis is better than the advanced composition theorem of DP? Does the discretization reintroduce DP accounting similar to the composition-style analysis? Can the authors further clarify how the resulting order is better than previous results?**
>   It is true that the results for the entire trajectory (Proposition 4.5, Theorem 5.4, and Theorem 5.8) essentially match those given by Rényi composition bounds. However, they exceed those given by the advanced $(\epsilon,\delta)$ composition bound and thus remove the need for complicated privacy accounting. We shall make this clearer in the revised version.
>   However, the results for the final draw from the Markov chain (Proposition 4.6, Theorem 5.2, and Theorem 5.6) are new and use novel probabilistic techniques. In particular, these results are uniform in time, which is never possible with composition-based analysis. Additionally, they apply to processes taking values on the whole space with non-convex assumptions. We are not aware of any work in the literature that considers this setting.

---

### Official Review · Reviewer_iGVP · 2025-03-14

**Overall Recommendation:** 2

**Summary:**

This is a theoretical work studying DP and MCMC algorithms, focusing on:
1. Connections between mixing, the privacy of the exact posterior, and the privacy of intermediate iterates.
1. The privacy of Markov chains based on Langevin diffusion, e.g. the Unadjusted Langevin algorithm (ULA).

I view the central new result as a uniform-in-time bound on the privacy loss of ULA in a certain non-convex loss function. Informally, we require the loss to be strongly convex outside a ball. The key step in the proof is to show that, for coupled versions of this process run on adjacent datasets, the iterates are never far apart.

**Claims And Evidence:**

Yes.

**Essential References Not Discussed:**

[1] study the privacy loss of Langevin diffusion. [2] give uniform-in-T bounds on the privacy loss of DP-SGD in a non-convex setting.

[1] Ganesh, Arun, Abhradeep Thakurta, and Jalaj Upadhyay. "Langevin diffusion: An almost universal algorithm for private euclidean (convex) optimization." arXiv preprint arXiv:2204.01585 (2022).

 [2] Asoodeh, Shahab, and Mario Diaz. "Privacy loss of noisy stochastic gradient descent might converge even for non-convex losses." arXiv preprint arXiv:2305.09903 (2023).

**Experimental Designs Or Analyses:**

n/a

**Methods And Evaluation Criteria:**

Yes.

**Other Comments Or Suggestions:**

none.

**Other Strengths And Weaknesses:**

none.

**Questions For Authors:**

Without answers to the following questions, I cannot recommend acceptance.
1. How does your proof strategy in Section 4 differ from that of [1, Section 1.1.1], for example? How does it differ from the approaches in Chourasia et al. (2021) or Ganesh et al. (2022)?
1. Can you sketch how your paper needs to change in light of the extra citations?
1. In particular, can you re-summarize how your technical contributions improve upon existing work?

[1] https://dpcourse.github.io/2021-spring/lecnotes-web/lec-09-gaussian.pdf

**Relation To Broader Scientific Literature:**

This submission has serious issues here. There are key references missing (see below) and, with the current version, some readers might walk away misunderstanding the connections to existing work.

Here are a few issues I observe.
1. The informal takeaways from Section 3 I view as obvious. It is not clear to me what value the quantitative versions add. (For example: "Any MCMC algorithm that is asymptotically exact will fail to be $(\varepsilon,\delta)$-differentially private for some $n$ when the posterior itself is not $\(\varepsilon,\delta)$-differentially private.")
1. Section 4 claims a "novel proof strategy to obtain DP guarantees of MCMC algorithms," but it seems to me to be the normal strategy for proving approximate DP. I ask a question below.
1. The submission claims to solve Open Problem 1.1 from Altschuler and Talwar (2022), but (i) the submission gives an upper bound for a specific family of distributions and (ii) the submission gives no lower bound.
1. The paper says "we improved on known composition bounds," but then say "this essentially matches bounds presented in Chourasia et al. (2021)."

**Theoretical Claims:**

I inspected parts of the theoretical arguments closely and believe them to be correct. Overall I think the theory is sound.

---

> ### Author Rebuttal · Authors · 2025-03-31
>
> We thank the reviewer for their careful comments, and particularly for the suggestion of relevant connected work in the literature. In response to specific questions, we have the following responses:
>
> - **The informal takeaways from Section 3 I view as obvious. It is not clear to me what value the quantitative versions add...**
>   The takeaways from Section 3 are indeed rather intuitive, but we did not find any such result in the literature. On the contrary, several papers proposed novel MCMC algorithms with DP-guarantees of the one-step transition kernel, motivating these as means to perform differentially private Bayesian inference. Our results from Section 3 (e.g., Proposition 3.4) make it very clear that such MCMC algorithms cannot achieve their goal of giving samples that are close to the posterior distribution unless the posterior distribution itself enjoys DP. This strongly encourages a shift in the approach to differentially private Bayesian inference.
>
> - **Section 4 claims a "novel proof strategy to obtain DP guarantees of MCMC algorithms," ...**
>   We shall make this aspect clearer in the next version of the paper, including a discussion on the following points. In Section 4.1, we give rigorous statements on how to obtain DP using the Radon-Nikodym (RN) derivative of the laws of the randomized algorithms for two adjacent datasets. This is much more general than the result in the notes the reviewer mentioned, but it is indeed not intrinsically a novel approach. However, thanks to our formal statement, it becomes clear that the DP of MCMC algorithms based on diffusions can be studied with Girsanov's theorem, which indeed gives an expression for the RN derivative of interest. Finally, we obtain results for the algorithm that releases only the $n$-th step of the chain with a novel perturbation technique, once again relying on Girsanov's theorem.
>
> - **The submission claims to solve Open Problem 1.1 from Altschuler and Talwar (2022), but (i) the submission gives an upper bound for a specific family of distributions and (ii) the submission gives no lower bound.**
>   It is true that we do not calculate exactly the optimal privacy parameters. We thank the reviewer for noting this, and we shall be more precise in the revised version. However, we do believe that we address Question 1.2 (*Does the privacy loss of Noisy-SGD increase ad infinitum in the number of iterations?*) and make progress towards Question 1.1 (*What is the privacy loss of Noisy-SGD as a function of the number of iterations?*). We note that optimal privacy bounds are only known in the Gaussian case (see Theorem 3 in Chourasia et al. (2021)) and are unlikely to be tractable in general.
>
> - **The paper says "we improved on known composition bounds," but then says "this essentially matches bounds presented in Chourasia et al. (2021)."**
>   Indeed, the improvement in composition bounds is only in comparison to known bounds for $(\varepsilon, \delta)$ privacy, and not in comparison to known Rényi composition bounds. We thank the reviewer for noting this and shall make this point clearer in the revised version. Chourasia et al. (2021) proves a similar result using PDE machinery.
>
> Regarding the Questions for authors section.
> The references provided by the reviewer are very interesting and explore pertinent connections. We are going to add them to the revised version of the paper, together with a discussion on the differences with our work.
>
>  For these references and every work in the literature of privacy of Markov chains we are aware of, one of the following holds:
> 1. The domain of the process is bounded.
> 2. The privacy bounds degenerate with the number of steps.
> 3. The deterministic part of the Markov kernel is contractive (i.e., the gradient of a strongly convex function).
>
> The main novelty of our work is that we prove privacy bounds without 1, 2, or 3, and use different and novel technical tools to do so. The tools used to prove privacy in [1] are either increasing bounds on Rényi divergence (see Lemma 2.2) or composition bounds that require noise that increases with time to be uniform (see Lemma 3.1).  In [2], all results are either on a bounded state space or degenerate with the number of steps (for large $T>0$, the RHS of (21) will be greater than $1$). Chourasia et al. (2021) considers a more similar setting and utilises a quite ingenious PDE argument to show uniform-in-time privacy bounds for a single draw from the SGD chain. However, in order to obtain explicit bounds, the authors have to assume strong convexity of the loss function. In comparison, we use a hybrid pathwise-perturbation approach, wherein we show that the two processes in question are almost surely close, then show the Radon-Nikodym derivative of their laws is mostly small by means of a perturbative argument and Girsanov's theorem. We are not aware of any such argument in the privacy literature.

---

> > ### Comment · Reviewer_iGVP · 2025-04-08
> >
> > Thank you for your response, this has clarified the main questions I had.

---

### Official Review · Reviewer_dztJ · 2025-03-19

**Overall Recommendation:** 4

**Summary:**

This paper present theoretical results on differential privacy guarantees for Markov Chain Monte Carlo (MCMC) methods. They develop DP guarantees for both full chains and for the final state of the chain, and demonstrate how these results can be applied for specific instances of MCMC dynamics.

## Update after rebuttal

The authors have answered my questions and I believe with the promised clarifications that the paper will be a valuable contribution.

**Claims And Evidence:**

Clear and convincing evidence for the proofs with which I could properly engage (all up to Appendix B.3), after that everything looks ok to me but I am less confident about given that I do not routinely use these theoretical tools.

**Essential References Not Discussed:**

None.

**Experimental Designs Or Analyses:**

Not applicable.

**Methods And Evaluation Criteria:**

Not applicable.

**Other Comments Or Suggestions:**

Here are what I think are some typos I spotted:

- In the line preceding Section 3, the divergence $D$ uses a comma rather than $\Vert$ to separate arguments, inconsistent with other instances of this divergence

- Section 3, 2nd paragraph: did you mean to say $(\epsilon, \delta + \beta(e^{\epsilon} + 1))$-DP of $\nu_D$?

- Final sentence of first paragraph of 4.2, and in the proofs in the Appendices, you write $(X_T^D)\_{t \in [0,T]}$ rather than $(X_t^D)_{t \in [0,T]}$ and I imagine the latter is what you want so that the index actually appears somewhere?

- Appendix B.1: final equality should have $\mathbb{P}$ not $\mathbb{E}$ on RHS

**Other Strengths And Weaknesses:**

I think the paper is clear and well-presented. The introduction to differential privacy and Renyi privacy was good and well-paced. I especially appreciate the efforts the authors went to to provide hyperlinks in both directions from propositions and proofs, allowing me to easily jump back and forth between statements and their proofs presented in the Appendix. To my knowledge, the work is original and in my view presents significant results at the intersection of MCMC and privacy guarantees.

Just a few comments on improving clarity:

- $P_{\mathcal{A}(\mathcal{D})}$ is used in Definition 2.2 on Page 2, but not defined until Page 5 (as far as I could see) – it was reasonably clear from context what it meant, but I think for improved clarity it would be beneficial to have defined it earlier.

- I didn't like the overloading of notation, e.g., $\eta$ is first a measure on $(E, \mathcal{B}(E))$ but then a real parameter appearing in the discussion the DP of Monte Carlo estimators

**Questions For Authors:**

**Conclusions**

Sorry for the basic question but what does it mean to "choose" a Bayesian posterior that has good DP properties? Do you just mean that you need to choose a model-prior combination that results in differentially private posteriors in order for any of the guarantees you provide to hold?

**Proposition A.3**

- What are $\mu$ and $\nu$ without subscripts?

- Should it be $\leq$ at bottom of page 11?

- Re. “Suppose now that $\vert{\vert{\mu_{\mathcal{D}'} - \nu_{\mathcal{D}'}\vert}\vert}\_{\text{TV}} \leq \zeta$ for some constant $0 < \zeta < \delta_{\mu}−\delta_{\nu}$.” – is this Assumption 3.1 in action?

- In the final inequality, where did the min over the two arguments come from?

**Assumption A.4**

What is $\tilde{\delta}$ and how does it relate to the $\delta$ that is assumed to exist in that Assumption?

**Appendix A.4**

$\delta$ in first factor of RHS of equation on top of page 13, shouldn’t this be $\tilde{\delta}$?

**Relation To Broader Scientific Literature:**

From my perspective the relation to the broader literature is strong. The authors connect well to both background literature on MCMC and privacy, so I do not see issues here.

**Theoretical Claims:**

I went through all the proofs in detail up until Appendix B.3. The proofs in Appendix B.3 and onwards I checked, but as stated above I am less confident about commenting on given that I do not routinely use these theoretical tools.

---

> ### Author Rebuttal · Authors · 2025-03-31
>
> We are grateful to the reviewer for the careful comments. In the revised version of the paper, we will incorporate all their comments and fix all the typos, notation clashes, and other issues that were mentioned by the reviewer. Below we reply to the questions that the reviewer raised:
>
> - **Sorry for the basic question but what does it mean to "choose" a Bayesian posterior that has good DP properties? Do you just mean that you need to choose a model-prior combination that results in differentially private posteriors in order for any of the guarantees you provide to hold?**
>
>     Indeed, we mean exactly what the reviewer wrote. We will clarify this aspect in the revised version of the paper.
>
> - **What are $\mu$ and $\nu$ without subscripts?**
>
>     We believe the reviewer refers to Proposition A.2. This was indeed an imprecise notation. We meant to refer to the families of probability distributions $\{\mu_{\mathcal{D}}:\mathcal{D}\in\mathcal{S}\}$, $\{\nu_{\mathcal{D}}:\mathcal{D}\in\mathcal{S}\}$. We will use the correct notation in the revised paper.
>
> - **Should it be $\leq$ at bottom of page 11?**
>
>     Here there was a typo before the inequality that the reviewer points to. Since we use the strict inequality on line 590, it follows we have strict inequalities also in lines 597-599. For this reason, we have a strict inequality also in the display at the bottom of page 11.
>
> - **Re. “Suppose now that $\lVert \mu_{\mathcal{D}'} - \nu_{\mathcal{D}'} \rVert_{TV}\leq \zeta$ for some constant $\zeta<\delta_\mu - \delta_\nu$.” – is this Assumption 3.1 in action?}, and also: \emph{In the final inequality, where did the min over the two arguments come from?**
>
>     In the revised version of the paper, we will clarify our proof strategy.
>     At this stage of the proof of Proposition A.3, we show that if $\lVert \mu_{\mathcal{D}'} - \nu_{\mathcal{D}'} \rVert_{TV}\leq \zeta$, then it must be that $\lVert \mu_{\mathcal{D}} - \nu_{\mathcal{D}} \rVert_{TV} > e^{-\varepsilon}(\delta_\mu-\delta_\nu-\zeta)$. Alternatively, $\lVert \mu_{\mathcal{D}'} - \nu_{\mathcal{D}'} \rVert_{TV} > \zeta$. Hence, in either case, the two distributions are "far" in TV distance either for $\mathcal{D}$ or for $\mathcal{D}'$.
>
>     We do not know whether $\lVert \mu_{\mathcal{D}'} - \nu_{\mathcal{D}'} \rVert_{TV} \leq \zeta$ or $\lVert \mu_{\mathcal{D}'} - \nu_{\mathcal{D}'} \rVert_{TV} > \zeta$, but regardless we know that for $\tilde {\mathcal{D}} \in \{\mathcal{D},\mathcal{D}'\}$ we have the lower bound shown in the equation on lines 613-614. By taking the minimum, we take the less stringent lower bound, which always holds. Finally, we optimize for $\zeta$ to obtain the largest lower bound. This is possible since $\zeta$ can be chosen arbitrarily.
>
> - **What is $\tilde \delta$ and how does it relate to the $\delta$ that is assumed to exist in that Assumption?**
>
>     This was a typo. In Assumption A.4, we should have written "There exist $\eta,\tilde{\delta}$ such that ..."
>
> - **In the first factor of the RHS of the equation on top of page 13, shouldn’t this be $\tilde\delta$?**
>
>     Indeed, we will correct this in the revised version.

---

> > ### Comment · Reviewer_dztJ · 2025-04-02
> >
> > Thank you to the authors for their response and for clarifying these things. I think the paper is good (with the promises the authors give to correct typos/clarifications in the revision) and I am happy to maintain my score.

---

### Decision · Program_Chairs · 2025-05-01

**Decision:**

Accept (poster)

**Comment:**

The reviewers agree that this is a well-written paper, proving and gathering useful theoretical results and tools to study the differential privacy properties of MCMC algorithms. This paper is theoretical with no empirical work, but I don't think this is a problem: all the theoretical results follow a clear narrative so that their correctness and validity seems reasonable, and there isn't an unreasonable amount of proofs stashed away in the appendix. The reviewers have provide some suggestions to improve this work, and the authors have offered some ways to incorporate these: please read the discussions again carefully while preparing the final version.

Some comments I have (in addition to those of the reviewers):
1. Reviewer iGVP mentioned some of the results in Section 3 are "obvious". I think they are still useful and the rest of the paper still has more than enough contributions. However, I wonder if the example cited: Corr 3.3: "there exists an n such that the MCMC output is not DP if the posterior is not DP" can be strengthened to "there exists an n_0 such for all n > n_0, the MCMC output is not DP if the posterior is not DP"?

2. Lemma 4.1: isn't this essentially the usual interpretation of (\epsilon,\delta)-DP with likelihood ratio replaced by Radon-Nikodym derivative. That is, \epsilon,0 DP with a \delta probability of failure is exactly (\epsilon,\delta)-DP (see e.g. Lemma 7.1.5 of Vadhan 2017, "Complexity of differential privacy"). I think something similar is probably true for Lemma 4.2. I think the RN-derivative framing is still useful for the applications with diffusions, but I'm not sure how novel these lemmas actually are, and I think these existing results should be elaborated in the paper:.